# Quantifying CH<sub>4</sub> point source emissions with airborne remote sensing: First results from AVIRIS-4

Sandro Meier<sup>1, 2</sup>, Marius Vögtli<sup>2</sup>, Andreas Hueni<sup>2</sup>, Audrey McManemin<sup>3</sup>, Adam R. Brandt<sup>3</sup>, Catherine Juéry<sup>4</sup>, Vincent Blandin<sup>5</sup>, Dominik Brunner<sup>1</sup>, and Gerrit Kuhlmann<sup>1</sup>

Correspondence: Sandro Meier (sandro.meier@empa.ch), Gerrit Kuhlmann (gerrit.kuhlmann@empa.ch)

Abstract. Atmospheric concentration of methane (CH<sub>4</sub>), a critical greenhouse gas, increased significantly since pre-industrial times, with anthropogenic emissions originating primarily from agriculture, fossil fuel use and waste management. However, considerable uncertainties persist in the detection and quantification of anthropogenic CH<sub>4</sub> emissions. In this study, we present first CH<sub>4</sub> observations, plume detections and emission estimates from the new state-of-the-art Airborne Visible InfraRed Imaging Spectrometer 4 (AVIRIS-4), which participated in a blind controlled release experiment in September 2024 in southern France. We used an albedo-corrected matched filter to retrieve CH<sub>4</sub> maps from the spectral images and estimated CH<sub>4</sub> emission with the Integrated Mass Enhancement (IME) and Cross-Sectional Flux (CSF) methods. Our results demonstrate that AVIRIS-4 can reliably detect emissions as low as 5.5 kg  $\mathrm{CH_4\,h^{-1}}$  under good weather conditions at low flight altitudes  $(<1500 \,\mathrm{m})$  and 1.45  $\,\mathrm{kg}\,\mathrm{CH_4}\,\mathrm{h^{-1}}$  under ideal conditions. While AVIRIS-4 provides highly accurate  $\mathrm{CH_4}$  maps at  $<0.5 \,\mathrm{m}$  resolution, emission estimation is limited by the accuracy of the effective wind speed, whose uncertainty and natural variability contribute substantially to the overall uncertainty. Using wind speed at source height performs well for small releases (below  $20 \text{ kg CH}_4 \text{ h}^{-1}$ ) (rRMSE = 1.065; rMBE = 0.361) and overall (rRMSE = 0.702; rMBE = -0.204). Using literature-derived effective wind speeds improves the apparent fit between estimated and reported CH<sub>4</sub> emissions, but degrades performance both in overall agreement (rRMSE = 2.098; rMBE = 0.964) and for low-emission events (rRMSE = 2.367; rMBE = 1.711). Interestingly, the high spatial resolution makes it possible to retrieve the cast shadow of the CH<sub>4</sub> plume, which can be used to estimate source and plume height, and could provide an approach for better constraining the height-dependency of the effective wind speed. On the bottom line, the controlled release experiment provides critical insights into the sensor's capabilities and guides further improvements to detect and quantify low intensity sources in the fossil fuel and waste management sectors, with implications for more accurate global greenhouse gas monitoring.

<sup>&</sup>lt;sup>1</sup>Empa, Laboratory for Air Pollution / Environmental Technology, Ueberlandstrasse 129, 8600 Duebendorf, Switzerland

<sup>&</sup>lt;sup>2</sup>Department of Geography, University of Zurich, Winterthurerstrasse 190, 8057 Zurich, Switzerland

<sup>&</sup>lt;sup>3</sup>Department of Energy Science & Engineering, Stanford University, United States

<sup>&</sup>lt;sup>4</sup>Air Quality Laboratory, TotalEnergies, France

<sup>&</sup>lt;sup>5</sup>TotalEnergies Anomalies Detection Initiatives (TADI), TotalEnergies, France

#### 1 Introduction

20

Methane (CH<sub>4</sub>), a potent greenhouse gas with a global warming potential 28 times higher than  $CO_2$  on a timescale of 100 years, has seen an almost threefold rise from 700 ppb pre-industrial levels to over 1900 ppb due to natural and anthropogenic sources (Seinfeld and Pandis, 2016). Major contributors include agriculture, fossil fuels, and waste. Due to its short lifetime of only 9 years,  $CH_4$  is removed more quickly compared to most other greenhouse gases. Reducing  $CH_4$  emissions is therefore considered an effective measure to mitigate anthropogenic climate change in the near term. However, there are still significant uncertainties in the quantification of anthropogenic  $CH_4$  emissions (Saunois et al., 2020).

For instance, Saunois et al. (2020) estimated that uncertainties in emissions from the fossil fuel sector are around 20-35% with strong regional variations. Reducing these uncertainties is challenging for several reasons. One of them is the fact that an important fraction of anthropogenic CH<sub>4</sub> emissions, e.g., from the fossil fuel sector, result from unintentional leakage, which cannot be accurately quantified. Additionally, global CH<sub>4</sub> emission estimates depend on a network of monitoring stations, which is dense and accurate in northern and mid-latitudes but sparser in other regions (Saunois et al., 2020). For these reasons, satellite remote sensing observations of CH<sub>4</sub> have been used to estimate the emissions in a top-down approach (e.g. Alexe et al., 2015; Bousquet et al., 2018; Fraser et al., 2013). These remote sensors can be separated into area flux mappers (e.g. Sentinel-5P, MethaneSAT, GOSAT-GW and CO2M) which are designed to have a global to regional coverage and point flux mappers (e.g. Landsat-8, Sentinel-2, GHGSat, PRISMA and EnMAP) which are used to observe regional to local emissions (Jacob et al., 2022).

Most of the currently available  $CH_4$  imagers are limited by spatial and/or spectral resolution which hinders the precision and accuracy of the emission estimates (Bousquet et al., 2018). This results in high detection limits in the range of a few 100 to several 1000 kg  $CH_4$  h<sup>-1</sup> for spaceborne instruments such as Sentinel-2 and Sentinel-5, PRISMA, EnMAP or GHGSat (e.g. Jacob et al., 2022; Gorroño et al., 2023; Joyce et al., 2023). For airborne instruments with a higher spatial resolution such as MethaneAIR, GHGSat-AV and the Airborne Visible InfraRed Imaging Spectrometer - Next Generation (AVIRIS-NG), the detection limit decreases to 10 to 100 kg  $CH_4$  h<sup>-1</sup> under favourable conditions (e.g. Cusworth et al., 2021; Duren et al., 2019; Jongaramrungruang et al., 2022; Kuhlmann et al., 2025; Guanter et al., 2025).

 ${
m CH_4}$  emissions from sources with small emission strengths that cannot be quantified from space (<  $100~{
m kg\,CH_4\,h^{-1}}$ ) are crucial for two reasons: First, leakages from the production and use of fossil fuels are often small and remain undetected by satellite-based approaches. Second,  ${
m CH_4}$  emissions from oil and gas production have a lognormal distribution with many small sources but only a few large ones (e.g. Balcombe et al., 2018; Stavropoulou et al., 2023; Williams et al., 2025). Accurate knowledge of the emission distribution of sources from a given sector or country is crucial for extrapolating  ${
m CH_4}$  emissions from the entire sector or country by accounting for sources below the detection limit (Zavala-Araiza et al., 2015; Zhang et al., 2023; Kuhlmann et al., 2025).

The detection of low intensity CH<sub>4</sub> sources requires a sensor that combines high spatial resolution with a good signal-tonoise ratio. One such state-of-the-art sensor is the new Airborne Visible InfraRed Imaging Spectrometer 4 (AVIRIS-4). It was developed by NASA JPL as a successor of AVIRIS-NG in parallel to its sister instruments Earth Surface Mineral Dust Source

Investigation (EMIT) and AVIRIS-3 which are in service on board the ISS and as airborne sensor respectively (Hueni et al., 2025). In this paper, we present the processing chain for retrieving CH<sub>4</sub> emissions from AVIRIS-4 measurements, show CH<sub>4</sub> maps and emission estimates from a blind controlled release experiment and characterise the capabilities and limitations of AVIRIS-4 for CH<sub>4</sub> emission quantification. The analysis considers the influence of flight altitude, meteorological conditions such as wind speeds and atmospheric stability, illumination and viewing conditions, and surface reflectance on the detection limit and the quality of the emission quantifications, providing guidance for future campaigns.

# 60 2 Data and Methods

This section covers the description of AVIRIS-4 (Section 2.1) used for the acquisition of remote sensing data in the controlled release experiment (Section 2.2) and the data processing chain from radiance data processing (Section 2.3), CH<sub>4</sub> retrieval (Section 2.4) and CH<sub>4</sub> emission estimation (Section 2.5) to the estimation of uncertainties (Section 2.6).

#### 2.1 AVIRIS-4 sensor specification

AVIRIS-4 is a state-of-the-art imaging spectrometer with identical core components as NASA JPL's AVIRIS-3 and the EMIT spectrometer (Green et al., 2022; Shaw et al., 2022; Hueni et al., 2025). The spectrometer is equipped with a 1280-pixel sensor array and records hyperspectral data in 328 bands spanning the ultraviolet (UV) to the shortwave infrared (SWIR). In practice, 1241 pixels receive sufficient illumination and SNR, and 287 bands are retained for data processing. Detailed sensor specifications are provided in Hueni et al. (2025). Compared to its predecessor it offers enhanced stability, spatial sampling interval (hereafter referred to as spatial resolution) and signal-to-noise ratio (SNR) (see Table 1).

## 2.2 Controlled release experiment

The data for this analysis was acquired during a single-blind controlled release experiment organised by the Environmental Assessment and Optimization Group at Stanford University between the  $16^{th}$  and  $20^{th}$  of September 2024 at the TotalEnergies Anomalies Detection Initiatives (TADI) site in Lacq in the south of France (latitude:  $43.412^{\circ}$ , longitude:  $-0.63643^{\circ}$ , elevation a.s.l.: 95 m) (see Figure 1a). A total of 13 commercial and academic teams, using a range of technologies - including continuous monitoring, vehicle-based measurements, drones, airborne in-situ measurements, remote sensing from aircraft, and satellites - participated in the experiment. The results of all teams were collected and analysed in McManemin (2025). On each campaign day (8:00 - 18:00 CEST), up to 9 individual controlled releases with rates varying between 0.02 and 350 kg CH<sub>4</sub> h<sup>-1</sup> were conducted at different unknown heights between 0.01 to 6.5 m above ground and at different unknown locations on the study site (see Figure 1b). Each release lasted for 45 minutes and was followed by a 15 minute break before the start of the next release. In some periods, no CH<sub>4</sub> was released to enable the detection of false positives. Additionally, the wind speed was measured using a ZX 300 Doppler wind lidar positioned 100 m from the emission sources. The instrument recorded horizontal and vertical wind speeds, as well as wind direction, at preselected heights between 10 and 300 m above ground level, with a temporal resolution of approximately 20 seconds. Participating teams were aware of the timing of releases while locations and

Table 1. Specifications of AVIRIS-4 compared to AVIRIS-NG, adapted from Green et al. (2022) and Hueni et al. (2025).

| Category                    | AVIRIS-4                   | AVIRIS-NG                  |  |
|-----------------------------|----------------------------|----------------------------|--|
| SPECTRAL                    |                            |                            |  |
| Range                       | 375 to 2504 nm             | 380 to 2510 nm             |  |
| Sampling                    | 7.4 nm                     | 5 nm                       |  |
| Response (FWHM)             | 1 to 1.5 $\times$ sampling | 1 to 1.5 $\times$ sampling |  |
| Calibration                 | $\pm 0.1~\mathrm{nm}$      | $\pm 0.1~\mathrm{nm}$      |  |
| RADIOMETRIC                 |                            |                            |  |
| Range                       | 0 to max Lambertian        | 0 to max Lambertian        |  |
| Signal-to-noise ratio (SNR) | >3000 @ 600 nm             | >2000 @ 600 nm             |  |
|                             | >1200 @ 2200 nm            | >1000 @ 2200 nm            |  |
| Calibration                 | 97% (<3% uncertainty)      | 95% (<5% uncertainty)      |  |
| SPATIAL                     |                            |                            |  |
| Swath samples               | 1241                       | 600                        |  |
| Swath angle                 | 40.2° field-of-view        | 34° field-of-view          |  |
| IFOV                        | 0.6 mrad                   | 1 mrad                     |  |
| FPS                         | 213                        | 10 - 100                   |  |
| Response (FWHM)             | 1 to 1.5 $\times$ sampling | 1 to 1.5 $\times$ sampling |  |

flow rates of the releases as well as the wind data were only made available after all teams had submitted their initial emission estimates. Details of the release experiment, the participating teams and the synthesis can be found in McManemin (2025). For the campaign, AVIRIS-4 was mounted on a hydraulic stabilisation mount and built into a Cessna 208B Grand Caravan EX. The aircraft flew over the release site in either north-south or east-west direction at different altitudes of 12000, 9000, 6000, 4200 and 3300 ft or 3660, 2740, 1830, 1280, 1000 m above mean sea level (amsl) (see Figure 1a). This resulted in average spatial resolutions of 2.0, 1.5, 1.0, 0.7 and 0.5 m across and 0.35 m along. For the remainder of the article, all wind speed heights are given in metres above ground level and all flight altitudes in feet above mean sea level.

# 2.3 Data processing

# 2.3.1 Radiometric and spectral calibration, georeferencing

The level 0 data acquired by AVIRIS-4 consists of raw digital numbers and has the spatial dimensions along-track and across-track as well as a spectral dimension. The level 0 data was converted into level 1 at-sensor radiances (in  $\mu$ W cm<sup>-2</sup> nm<sup>-1</sup> sr<sup>-1</sup>) using laboratory-measured calibration coefficients. The level 1 data was georeferenced using a parametric approach (Schläpfer and Richter, 2002), where the geometry of the sensor, its location and orientation acquired from GNSS and inertial navigation

**Figure 1.** (a) Location of the controlled release experiment at the TADI site in the south of France (red polygon in the inset map). Superimposed are the imaging footprints of AVIRIS-4 for overpasses at 3300 and 12000 ft. (b) Aerial view of the site with the locations of the wind lidar as well as the potential release locations. The background maps are obtained from © Google Earth.

system (INS) data were combined with a digital elevation model (IGN, 2018) to project the radiometrically corrected data onto the surface with sub-pixel accuracy. Details on the processing are described in Hueni et al. (2025).

# 100 2.3.2 Masking of shadows and water surfaces

105

Dark surfaces such as cast shadows and water bodies have a low SNR and therefore produce artefacts when processing the data. Additionally, cast shadows only contain diffuse radiance, which is inconsistent with the non-scattering assumption in CH<sub>4</sub> retrieval. Cast shadows were especially pronounced in our data, as the controlled release experiment took place in late September under low solar zenith angles (SZA). For this reason, we masked these areas using a modified version of the cast detection method described in Schläpfer et al. (2018), using radiances at 450 nm for blue ( $L_b$ ), 670 nm for red ( $L_r$ ) and 780 nm for near-infrared (NIR) ( $L_n$ ):

$$i_{sh} = \frac{L_r + k_n (L_n - L_r)_{>0}}{L_b} \cdot \frac{1}{a e^{b L_{b,dark}}}.$$
 (1)

We used the default parameters  $k_n = 0.1$ , a = 1.58 and b = -0.04 from Schläpfer et al. (2018). Next, we divided the inverse of the resulting index by the integrated radiance over all wavelengths. After empirical evaluation, values larger than 0.25 were masked prior to applying the matched filter.

# 2.4 CH<sub>4</sub> retrieval

We retrieved CH<sub>4</sub> maps from the AVIRIS-4 radiance cube using the computationally efficient matched filter approach following Foote et al. (2020) and further refined by Kuhlmann et al. (2025). The filter detects a known signal within a noisy background by enhancing the signal relative to the noise, effectively maximising the output signal-to-noise ratio under the assumption of additive Gaussian noise.

#### 2.4.1 Matched filter

Using a linearised form of the Beer-Lambert law, the matched filter (MF) takes the form

$$\alpha_{\epsilon} = \frac{(L_{\text{obs}} - \hat{\mu}) \cdot \hat{S}^{-1} \cdot t}{t^{\top} \cdot \hat{S}^{-1} \cdot t}$$
(2)

where  $\alpha_{\epsilon}$  represents the CH<sub>4</sub> column enhancement,  $L_{obs}$  the observed spectrum in the two wavelength ranges 1480 to 1800 and 2080 to 2500 nm,  $\hat{\mu}$  and  $\hat{S}$  the median and covariance of the observed spectrum and  $t = \hat{\mu} \cdot -s$  the target spectrum. We used the negative of the unit absorption spectrum of CH<sub>4</sub> s to align Eq. 2 with derivations in other studies. Thereby, s is calculated using the radiative transfer equation assuming a geometric air mass factor (AMF), no atmospheric scattering according to Kuhlmann et al. (2025) and a CH<sub>4</sub> enhancement  $\epsilon$  in the lowest 1000-m layer respectively. The calculation of the plume-specific enhancement was achieved through an iterative approach, wherein the CH<sub>4</sub> maps were initially derived under the assumption of an enhancement of 0.01 ppm. The mean enhancement in the detected plume was then used for the subsequent iteration of the matched filter, which converged after two to three iterations.

# 2.4.2 Lognormal matched filter

Due to the linearisation of the Beer-Lambert law used in the derivation of the matched filter, it is only valid for weak CH<sub>4</sub> enhancements. Therefore, Schaum (2021) argued that a lognormal matched filter (LMF) provides the uniform most powerful solution for the detection of trace gas plumes, which takes the following form:

$$\alpha_{\epsilon} = \frac{(\ln(L_{\text{obs}}) - \hat{\mu}) \cdot \hat{\mathbf{S}}^{-1} \cdot \mathbf{s}}{\mathbf{s}^{\top} \cdot \hat{\mathbf{S}}^{-1} \cdot \mathbf{s}}$$
(3)

where s is the same unit absorption spectrum as above. This approach has been implemented and evaluated by Pei et al. (2023) for synthetic WRF-LES and observed data from the PRISMA satellite. According to Schaum (2021), the LMF could improve the detection performance for pixels with attenuated signal, e.g. with weak enhancement or at higher flight altitudes, due to the more realistic mean spectrum  $\hat{\mu}$  in logarithmic space.

# 2.4.3 Albedo correction

130

135

We applied the matched filter to the at-sensor radiance of each across-track position to avoid striping caused by differences in radiometric and spectral calibration of the sensor pixels. However, as outlined in Fahlen et al. (2024), the assumption that the

reference solar spectrum  $L_0$  can be approximated by the mean spectrum  $\hat{\mu}$  introduces a bias in the CH<sub>4</sub> column enhancement  $\alpha_{\epsilon}$  over heterogeneous surfaces, which must be corrected as follows:

$$\alpha_{corr} = \frac{1}{R} * \alpha_{\epsilon} \quad \text{with} \quad R = \frac{(L_{\text{obs}} * - s) \cdot \hat{S}^{-1} \cdot t}{t^{\top} \cdot \hat{S}^{-1} \cdot t} \tag{4}$$

While the correction factor helps mitigate biases in CH<sub>4</sub> enhancements, it also amplifies retrieval noise for dark surfaces with a low signal-to-noise ratio. This effect could be mitigated by masking cast shadows and water surfaces before applying the matched filter.

#### 145 2.4.4 Plume shadow correction

In some of the AVIRIS-4 observations, we observed double plumes due to plume shadows (see Sect. 3.5.4). They present a challenge for emission estimation because the  $CH_4$  retrieval assumes that the light traverses the plume twice, assuming a geometric AMF that depends both on the solar zenith angle (SZA) and viewing zenith angle (VZA):

$$AMF_{geom} = sec(SZA) + sec(VZA)$$
(5)

At the source location, however, the signal originating from the plume does not pass through the plume a second time after ground reflection, and thus it is independent of the VZA. Consequently,  $CH_4$  enhancements should be scaled by  $c_{plume}$ :

$$c_{\text{plume}} = \frac{\text{AMF}_{\text{geom}}}{\text{sec(VZA)}} \tag{6}$$

For the plume shadow enhancement, the respective correction factor is given by

$$c_{\text{shadow}} = \frac{\text{AMF}_{\text{geom}}}{\text{sec(SZA)}} \tag{7}$$

When the plume and its shadow were clearly resolved, we estimated the emissions and applied the corresponding correction factor. However, when the plumes partially overlapped, this separation was not feasible, limiting the applicability of the correction method. In such situations, we employed the integrated mass enhancement (IME), which aggregates all detected pixels without explicitly distinguishing between the plume and its shadow.

# 2.5 CH<sub>4</sub> emission estimation

- To estimate the CH<sub>4</sub> emissions, we used the integrated mass enhancement (IME) and cross-sectional flux (CSF) method implemented in the Python library for data-driven emission quantification (*ddeq*) (Kuhlmann et al., 2024). The CSF performs better for longer plumes and more turbulent conditions as it averages the fluxes along several cross-sections. Conversely, the IME is more appropriate for short plumes and plumes that deviate from a Gaussian plume shape. Both methods assume steady-state conditions of wind speed and emission rate. Limits of this assumption are further discussed in Section 4.2.
- All mass-balance based methods require an estimate of the wind speed U. Ideally, U would correspond to the effective wind speed  $U_{\rm eff}$ , which is the mean speed at which the plume is transported (Kuhlmann et al., 2024). However, as the vertical  $CH_4$  profile is unknown, we used four different approaches to obtain a wind speed estimate:


- 1. 10 m wind speeds  $U_{10}$  from ERA5 reanalysis data (Hersbach et al., 2018) as used for the initial reporting in McManemin (2025) as ground-based lidar measurements were not available prior to unblinding.
- 170 2. 10 m wind speeds  $U_{10}$  from wind lidar measurements.
  - 3. A linear scaling of the 10-m wind speed derived from model simulations for GHGSat (Varon et al., 2018):

$$U_{\text{eff}} = 1.47 \cdot U_{10}$$
 (8)

4. Wind speed at source height  $U_s$ , assuming a logarithmic wind profile (Fleagle and Businger, 1980; Seinfeld and Pandis, 2016). Wind profiles were derived assuming a surface roughness of 0.1 m and using on-site measurements of temperature and wind speed, combined with sensible heat fluxes from ERA5 reanalysis data (Hersbach et al., 2018). While plume rise and vertical mixing were not explicitly incorporated into the wind speed calculations, their potential influence was accounted for in the uncertainty analysis.

We also conducted a sensitivity analysis using lidar wind speeds at other elevations above ground level.

# 2.5.1 Integrated mass enhancement

The IME approach derives the emission rate Q based on the integrated mass enhancement M of a plume and a residence time  $\tau$  during which  $CH_4$  resides within the detectable plume. This residence time is approximated by the wind speed U and the length L of the detectable plume (Kuhlmann et al., 2024).

$$Q = \tau \cdot M = \frac{U}{L} \cdot M \tag{9}$$

The plume length L was calculated as the arc length of the centre line curve fitted to the detected plume.

The integrated mass M was computed as

$$M = \sum_{(i,j)\in\mathcal{P}_a}^{n} (V_{i,j} - V_{\text{bg}}) \cdot A_{i,j}$$
(10)

where  $V_{i,j}$  is the vertical column density,  $V_{\rm bg}$  is the background vertical column density, and  $A_{i,j}$  is the pixel area. The trace gas mass was summed up over the n pixels of the integration area  $\mathcal{P}_a$  which was obtained by a sufficient extension of the detected plume in the crosswind direction to include pixels with enhancements below the detection limit. A local CH<sub>4</sub> background  $V_{\rm bg}$  was calculated by applying a low-pass Gaussian filter to the CH<sub>4</sub> maps after masking the enhancements including a buffer (Kuhlmann et al., 2024).

# 2.5.2 Cross-sectional flux method

For the CSF, the detected plume is divided into multiple polygons. As in Kuhlmann et al. (2024), a Gaussian curve with linear background trend was then fitted to the  $CH_4$  enhancements of each polygon to obtain the line densities q:

$$g(y) = \frac{q}{\sqrt{2\pi}\sigma} \exp\left(-\frac{(y-\mu)^2}{2\sigma^2}\right) + my + b \tag{11}$$

Here, y is the across-plume direction,  $\sigma$  to the standard width and  $\mu$  to the mean of the fitted Gaussian curve with linearly changing background with slope m and offset b.

The emissions Q were then calculated as the product of the wind speed U and the uncertainty-weighted mean of all line densities  $\bar{q}$ :

$$Q = U \cdot \bar{q} \tag{12}$$

# 2.6 Estimation of uncertainty

Below we describe how uncertainty components are estimated and propagated for each input to the emission quantification.

# 2.6.1 CH<sub>4</sub> Columns

The uncertainty of  $CH_4$  columns  $\sigma_V$  was calculated as

$$\sigma_V = \sqrt{\sigma_t^2 + \sigma_{inst}^2} \tag{13}$$

where  $\sigma_{\rm inst}$  accounts for correlated and uncorrelated radiance-dependent instrument uncertainties. For this analysis, it was calculated using the EMIT noise model and propagated through the MF according to Fahlen et al. (2024). For future analyses, a noise model for AVIRIS-4 will be used, which is currently under development.  $\sigma_{\rm t}$  represents correlated uncertainties in the target t due no-scatter assumptions for the calculation of the unit absorption spectrum s, which was estimated at a conservative 5% for this campaign based on Kuhlmann et al. (2025).

#### 2.6.2 Pixel area



Uncertainty in pixel area ( $\sigma_A$ ) is treated as a systematic spatial uncertainty, reflecting geolocation and georectification errors. During this campaign, geolocation accuracy was reduced due to a faulty cable, which impaired the temporal synchronization between GNSS data and AVIRIS-4 measurements. To assess the resulting geolocation uncertainty, AVIRIS-4 imagery was visually compared with Google Earth reference imagery. Based on this comparison, a conservative uncertainty of 5% of the nominal pixel area was assumed. The cable issue has since been resolved, and additional measures have been implemented to prevent similar problems in future campaigns.

# 2.6.3 Wind speed

The uncertainty of the on-site measured wind speed  $\sigma_U$  is assumed to consist of four terms:

$$\sigma_U = \sqrt{\sigma_{\text{inst}}^2 + \sigma_{\text{rep}}^2 + \sigma_{\text{eff}}^2 + \sigma_{\text{var}}^2}$$
 (14)

The term  $\sigma_{\rm inst}$  represents the systematic measurement uncertainty of the wind lidar which was estimated as 5% of the wind speed, based on guidance from the site operators. The term  $\sigma_{\rm rep}$  represents the error associated with the spatial displacement between the wind lidar and the actual plume locations. Given the close proximity of the lidar to the source positions in this



study, this component is assumed to be negligible or already captured in  $\sigma_{\rm var}$  (see below). The term  $\sigma_{\rm eff}$  reflects the uncertainty introduced by the use of the wind speed at source height instead of a concentration weighted wind profile. It was quantified by calculating the mean relative difference between the wind speed at source height and a Gaussian-weighted logarithmic wind profile. For the latter, we weighted the logarithmic wind profile with Gaussian curves around the source height with standard deviations ranging from 0.1 to 5 m and source heights between 0.01 and 6.5 m as experienced during the controlled release experiment.  $\sigma_{\rm eff}$  was found to be in the order of 30% for sources between 0 and 1.5 m above the ground and less than 5% for sources which are more elevated. Here, we used an estimate of 15%. Finally,  $\sigma_{\rm var}$  represents the uncorrelated errors due to the natural variability of on-site measured wind data during the overpass. It was quantified as the standard deviation of  $U_{10}$  over a one-minute window, consistent with the typical residence time of most detectable plumes, which was estimated to be no more than one minute.

#### 2.6.4 IME

The uncertainties of the emission estimates  $\sigma_Q$  of the IME were determined by the propagation of error:

$$\sigma_Q = Q \cdot \sqrt{\left(\frac{\sigma_U}{U}\right)^2 + \left(\frac{\sigma_L}{L}\right)^2 + \left(\frac{\sigma_M}{M}\right)^2} \tag{15}$$

The uncertainty of the plume length  $\sigma_L$  was estimated as 10% of the plume length or at least half of a pixel. The uncertainty of the integrated mass  $\sigma_M$  was calculated as

$$\sigma_{M} = \sqrt{\sum_{(i,j)\in\mathcal{P}_{a}}^{n} \left[ (A_{i,j} \cdot \sigma_{V_{i,j}})^{2} + (A_{i,j} \cdot \sigma_{V_{\text{bg}}})^{2} + ((V_{i,j} - V_{\text{bg}}) \cdot \sigma_{A_{i,j}})^{2} \right]}$$
(16)

where  $\sigma_{V_{i,j}}$  corresponds to the pixel-wise uncertainty of the vertical column density V. Using the trace gas column enhancement  $V_{\text{enh}_{i,j}} = V_{i,j} - V_{\text{bg}}$ , Eq. 16 simplifies to

$$\sigma_{\text{enh}} = \sqrt{\sum_{(i,j)\in\mathcal{P}_a}^n \left[ (A_{i,j} \cdot \sigma_{V_{\text{enh}_{i,j}}})^2 + (V_{\text{enh}_{i,j}} \cdot \sigma_{A_{i,j}})^2 \right]}$$

$$(17)$$

where  $\sigma_{A_{i,j}} = \sigma_{\bar{A}}$  and  $\sigma_{V_{\text{enh}_{i,j}}} = \sigma_{V_{\text{enh}}}$  were assumed to be constant and correspond to the mean within the plume.

#### 2.6.5 CSF

The uncertainties of the emission estimates  $\sigma_Q$  of the CSF were determined as

$$\sigma_Q = \sqrt{\bar{q}^2 \cdot \sigma_U^2 + U^2 \cdot \sigma_{\bar{q}}^2} \tag{18}$$

The uncertainty of the mean line densities  $\sigma_{\bar{q}}$  was obtained as the uncertainty of the mean of the fitted fluxes q along the plume, which accounts for uncertainties  $\sigma_q$  of the individual cross sections. Since  $\sigma_{\bar{q}}$  decreases with the square root of the number of line densities and does not account for the correlation of consecutive line densities, this uncertainty is set to at least 10% of the

mean line density:

$$\sigma_{\bar{q}} = \min(\sigma_{\bar{q}}, 0.1 \cdot \bar{q}) \tag{19}$$

The uncertainty of each cross-section  $\sigma_q$  was calculated from the uncertainty of the Gaussian fit to each sub-polygon  $\sigma_{\rm gauss}$  and the mean uncertainty of the pixel area  $\sigma_{\bar{A}}$  within a sub-polygon:

$$\sigma_q = \sqrt{\sigma_{\text{gauss}}^2 + \left(\frac{q}{\bar{A}} \cdot \sigma_{\bar{A}}\right)^2} \tag{20}$$

# 255 2.7 Probability of detection

We computed the probability of detection (POD) for AVIRIS-4 according to Conrad et al. (2023) as a function of reported emissions  $Q_{\text{rep}}$ ,  $U_{10}$  and flight altitude  $\tilde{h}$  using the flags "detected" and "not detected" by optimising the predictor and inverse link functions. This resulted in the following POD function:

$$POD = 1 - \left(1 + \left(1.03 \times 10^{10}\right) \frac{\left(5.18 \times 10^{8}\right) Q^{1.93}}{\left(\frac{\tilde{h}}{1000}\right)^{3.88} \left(u_{10} + 97.0\right)^{9.97}}\right)^{-1.84}$$
(21)

#### 260 3 Results

In what follows, we summarise the observing conditions relevant to  $CH_4$  retrievals during the controlled-release experiment (Section 3.1). We then present representative plume images from multiple releases across varied conditions (Section 3.2). Next, we assess how key parameters influence retrieval performance (Section 3.5), derive detection limits (Section 3.3), and compare estimated emissions with reported values (Section 3.4).

# 265 3.1 Controlled release experiment

In contrast to previous efforts, this new generation of controlled release experiments was planned to reflect more realistic natural conditions. While this allows to assess sensor performance in diverse terrain and meteorological conditions it also introduce limitations associated to different surface coverage, cast shadows and cloud conditions (see Figure 2 for detailed meteorological setting during all experiments, and Figures A3, A4, A5 and A6 in the Appendix for wind information). Despite these challenges, we were able to fly 100 overpasses at different hours of the day (see Figure 3 a) and at five altitudes (see Figure 3 b), which allowed us to evaluate the influence of wind speeds and spatial resolution on the  $CH_4$  detection and emission estimation. Flights at all flight levels were only scheduled for the first release in the morning and afternoon after refuelling. The atmospheric stability was estimated to be neutral to unstable for all observations based on the Pasquill stability classes using  $U_{10}$ .

Cloud cover ⊙: clear, ⊙: few, ⊙: scattered, ⊙: broken, ⊙: overcast

|                    |       | 16.09.2024 | 17.09.2024   | 18.09.2024   | 19.09.2024   | 20.09.2024 | SZA [°]<br>50 60 70 80 |
|--------------------|-------|------------|--------------|--------------|--------------|------------|------------------------|
| Local time [UTC+2] | 09:00 | 0          | n=0 ●        | n=0 ●        | n=0 ●        | n=0 ●      |                        |
|                    | 10:00 | 0          | n=0 ●        | n=0 ●        | n=0 ●        | n=0 ●      |                        |
|                    | 11:00 | n=5 🔾      | n=0 ●        | n=7 🐧        | n=0 ●        | n=0 ●      |                        |
|                    | 12:00 | n=8 🔾      | n=8 <b>①</b> | n=8 🔾        | n=0 ●        | n=0 ●      |                        |
|                    | 13:00 | n=8 🔾      | n=1 •        | n=7 🔾        | n=0 ●        | n=0 ●      |                        |
|                    | 14:00 | n=0 🔾      | n=0 <b>●</b> | n=4 O        | n=0 <b>●</b> | n=0 ●      |                        |
|                    | 15:00 | n=7 👁      | n=0 <b>④</b> | n=0 <b>⊙</b> | n=8 <b>①</b> | n=0 ●      |                        |
|                    | 16:00 | n=8 👁      | n=0 <b>④</b> | n=8 <b>●</b> | n=8 <b>⊙</b> | n=0 ●      |                        |
|                    | 17:00 | •          | •            | n=2          | n=8 🔾        | •          |                        |

Figure 2. Schedule of the controlled release experiment with the number of overpasses n for each release and a symbol for the average cloud conditions during the release. As the releases started either at '00, '30 or '45, the row label indicates the hour of the release end in local time. If no number of overpasses is given, no release took place during that time window. Bold entries indicate releases observed at all altitude levels; otherwise, observations were limited to 4200 and 3300 ft. The right-hand panel shows the average SZA for each hour.

Figure 3. (a) Number of overpasses at five different altitudes above mean sea level. (b) Number of overpasses at different hours of the day.


# 3.2 Examples of plume images

Figure 4 (upper row) presents three optimal examples of plumes resulting from three different releases. The plumes appear largely linear, with minimal influence from turbulence, which is favourable for emission estimation. For stronger sources, retrieval noise is barely noticeable, but at lower intensities - such as the  $26.4 \text{ kg CH}_4 \text{ h}^{-1}$  release - it can interfere with the plume signal and hinder accurate attribution of enhanced pixels (see Section 3.5.5).

Figure 4. (upper row) Linear  $CH_4$  plumes from release events with 26.4, 56.7 and 290 kg  $CH_4$  h<sup>-1</sup>, observed at 3300 ft at an average spatial resolution of 0.40 to 0.43 m. (lower row) Turbulent  $CH_4$  plumes from release events with 290 and 80.1 kg  $CH_4$  h<sup>-1</sup>, observed at 4200 ft at an average spatial resolution of 0.48 m.

The lower row in Figure 4 shows three turbulent plumes observed during overpasses at 4200 ft, where local enhancements caused by turbulent eddies are clearly visible. In these cases, the CSF method outperforms the IME approach, as the effect of turbulence is reduced through averaging across multiple cross-sections.

In addition to challenging conditions, there was also a case where turbulence impeded emission estimation, shown in Figure 5. A change in wind direction prior to the overpass appears to have caused a large, dispersed "blob" of  $CH_4$  enhancements. Since these conditions violate the steady-state assumption, this case was excluded from emission estimation.

# 3.3 Detection limit



The median noise level of  $CH_4$  maps was estimated to be around 700 ppm-m or 0.5 g  $CH_4$  h<sup>-1</sup> for the data of the controlled release experiment. Out of 100 overpasses, plumes were detected on 68 instances (Figure 6). In the most favourable case,

Figure 5.  $CH_4$  plume from release events with 52.94 kg  $CH_4$  h<sup>-1</sup>, observed at 3300 ft at a spatial resolution of 0.42 m.

the smallest observed plume corresponded to a  $1.45~\rm kg~CH_4~h^{-1}$  release at  $U_{10}=0.76~\rm m\,s^{-1}$  and a flight altitude of 4200 ft, representing the best-case detection limit for AVIRIS-4. Under typical conditions, plumes from releases of  $5.5~\rm kg~CH_4~h^{-1}$  and above were consistently detected at altitudes  $\leq 4200~\rm ft$ , with the exception of two overpasses where shadows from surface infrastructure obscured the signal. At higher flight altitudes ( $6000~\rm -12000~\rm ft$ ), detection performance was more constrained: for release rates  $\leq 9.23~\rm kg~CH4~h^{-1}$ , only one plume was detected ( $6000~\rm ft$ ,  $U_{10}=1.3~\rm m\,s^{-1}$ ), while the others could not be observed due to the combined effect of higher winds and lower emissions. The original objective of conducting observations

Figure 6. (a) Reported CH<sub>4</sub> emissions vs. on-site lidar wind measurement at 10 m. (b) Probability of detection for a flight altitude of 1000 m.

at multiple flight altitudes was to determine an altitude-dependent detection limit. However, because the  $CH_4$  release rates were not known in advance, the largest release event captured at all five altitudes was metered at only 9.23 kg  $CH_4$  h<sup>-1</sup>. This emission rate was below the detection threshold at altitudes above 6000 ft and therefore remained undetectable in those overpasses. As a consequence, no altitude-dependent detection limit could be established.



# 3.4 CH<sub>4</sub> emission estimation

Figure 7. Comparison between reported and estimated CH<sub>4</sub> emissions using (a) ERA5 10 m wind speeds, (b) lidar 10 m wind speeds, (c) effective wind speeds using  $1.4 \times u_{10}$  according to Varon et al. (2018) and (d) effective wind speeds at source height as described in Section 2.5. Insets enlarge the low-emission range and have an independent fit to the emission estimates. It is important to note that the  $R^2$  value represents the coefficient of determination of the weighted regression, which can take negative values.

We were able to estimate the emission from 67 of the 68 detected plumes, 54 of which were estimated using the CSF method and 13 using the IME method. Figure 7 shows the reported versus estimated  $CH_4$  emissions using four different wind speed inputs. As outlined in Section 2.5, the initial  $CH_4$  emission estimates were calculated using ERA5  $U_{10}$ , shown in subplot (a) of Figure 7. This approach yields a relatively weak correlation, with a fitted slope of only 0.53 and an  $R^2$  value of 0.55. Replacing ERA5 data with lidar-measured  $U_{10}$  in subplot (b) of Figure 7 substantially improves the agreement, increasing the slope to 0.65 and an  $R^2$  value of 0.73. This highlights the limitations of reanalysis wind data for accurate emission quantification (further shown in Figure A1). As a result, the use of ERA5 introduces both correlated and uncorrelated uncertainties in emission estimates that are difficult to quantify or correct.

Even when using on-site lidar wind speeds (Figure 7b), biases remain: emission rates for small release events tend to be overestimated, while large releases (e.g. at 80.1 and 290 kg  $\rm CH_4\,h^{-1}$ ) are significantly underestimated. This behaviour can be explained by plume dynamics: Small release events result in short plumes which remain near the emission height (





greater vertical mixing. The actual effective transport height may thus be above 10 m, resulting in an underestimation of emissions when using  $U_{10}$ . Additional influencing factors are specific to the release equipment, such as the outlet ejection velocity and whether the emission was oriented horizontally or vertically.

These limitations highlight the importance of estimating an effective wind speed ( $U_{\rm eff}$ ) that accounts for both source height and vertical mixing. Subplots (c) and (d) in Figure 7 compare two approaches: the method of Varon et al. (2018), which accounts only for vertical mixing, and the method developed in this study, which accounts only for source height. In subplot (c), the overall fitted trend lies close to the 1:1 line, but the estimates for small releases are substantially worse than when using  $U_{10}$ . This reflects the fact that Varon et al. (2018) derived the linear relationship between  $U_{10}$  and  $U_{\rm eff}$  for GHGSat, which has a coarser spatial resolution (50×50 m). At that scale, plumes have more time to mix vertically and are therefore transported by winds stronger than  $U_{10}$ . In contrast, subplot (d) shows a poorer overall trend than (c) due to the strong influence of large release events, but the estimates for small releases improve considerably. This suggests that short plumes are well captured because they remain close to the emission height, whereas vertical mixing is insufficiently accounted for in the case of larger releases.

To further investigate this hypothesis of strong vertical mixing, we incorporated lidar wind speeds at 20 and 38 m and calculated the uncertainty-weighted root mean squared error (RMSE) and relative mean bias error (MBE) between estimated and reported  $CH_4$  emissions, as shown in Figure 8. The results confirm that using  $U_{\rm src}$  substantially improves emission estimates for low intensity release events. For release events above  $30 \, \rm kg \, CH_4 \, h^{-1}$ , however, using  $U_{\rm src}$  tends to underestimate emissions and performs worse than estimates based on  $U_{10}$ ,  $U_{20}$  and  $U_{38}$ . Although the relative MBE decreases for larger releases, the high relative RMSE indicates substantial variability around the true values. This pattern may reflect the greater influence of turbulence on longer plumes compared to shorter ones.

**Figure 8.** Mean relative root mean squared error (RMSE) and relative mean bias error (MBE) between estimated and reported CH<sub>4</sub> emissions across emission bins.





In addition to source strength and therefore plume length, absolute wind speed appears to significantly influence the accuracy of emission estimates. This is illustrated in Figure 9, which shows the scaling factor required to align estimated emissions with reported values as a function of (a) plume length and (b) effective wind speed. While subplot (a) of Figure 9 supports the previously discussed hypothesis regarding plume length, subplot (b) reveals that lower wind speeds are associated with larger and more variable scaling factors. This observation aligns with the findings of Varon et al. (2018); Sánchez-García et al. (2022); McManemin (2025), who reported reduced accuracy in emission estimates across various techniques under low wind speed conditions. This is likely due to the increased variability typically observed at lower wind speeds. In contrast, we did not observe larger scaling factors for larger coefficients of variation (CoV) in wind direction in subplot (c) of Figure 9 as discussed in McManemin (2025). The reason for this is that our method does not depend on wind direction, as we do a nearly instantaneous measurement. The large spread in angles between the wind direction and the curve fitted to the plume in subplot (d) further highlights the strong influence of wind turbulence on the observed plumes.

**Figure 9.** Correlation of (a) plume length, (b) 10 m wind speed, (c) Coefficient of Variation (CoV) and (d) angle between plume curve and wind direction with the scaling factor required to align estimated CH<sub>4</sub> emissions using Lidar 10 m wind speeds with reported values.

This hypothesis is further supported by individual cases where estimated emissions diverge from reported values, as illustrated in Figure 10. Subplot (d) shows that the CH<sub>4</sub> fluxes across different cross-sections fluctuate strongly between 100 and 200 kg CH<sub>4</sub> h<sup>-1</sup> due to turbulent wind, likely reflecting both temporal variability in wind speed and changes in plume height that exposed it to different wind regimes. In such cases, one might consider using only CH<sub>4</sub> enhancements close to the source, such as those from the first cross-section, where  $U_{\rm eff}$  is expected to better approximate the wind speed at source height. However, this example shows that even this approach leads to underestimation, indicating that the measured wind speeds do not accurately reflect actual wind conditions. An analysis of the wind speed during the two minutes of the overpass reveals that  $U_{10}$  varies between 1 and 3 m s<sup>-1</sup>. For comparison, a 20 m plume under a 1 m s<sup>-1</sup> wind has a residence time of about 20 s, which matches the sampling interval of the wind lidar. As a result, the wind speed fluctuations visible in the plume cannot be resolved by the lidar, and the underestimation can likely be attributed to larger-than-expected temporal variability that is not captured at the instrument's temporal resolution.

**Figure 10.** (a) RGB image of the release location, (b) CH<sub>4</sub> map showing the detected plume and 12 cross-sections, (c) Gaussian fits to the CH<sub>4</sub> columns from the first and last three cross-sections, (d) along-plume flux of all cross-sections and retrieval metadata.

The analysis of uncertainty contributions to total emission estimate uncertainty (Figure A2) indicates that wind speed is
the dominant factor for both the CSF and IME methods. Most of this contribution arises from the natural variability of wind speed, with additional influence from uncertainty in the effective wind speed. In comparison, measurement errors in wind speed account for only a minor portion of the overall uncertainty.

Lastly, one source of deviation between estimated and reported  $CH_4$  emissions is the presence of cloud shadows over the release site as shown in Section 3.5.3, leading to the strong underestimations of the 80.1 kg  $CH_4$  h<sup>-1</sup> release event observed in Figure 7. Despite this underestimation, the plumes were still reliably detected, indicating that observations under suboptimal cloud conditions can still be valuable e.g. for leak detection.

# 3.5 Factors affecting the CH<sub>4</sub> retrievals

# 3.5.1 Spatial resolution



Figure 11 shows examples of AVIRIS-4 RGB images and CH<sub>4</sub> maps of the release site acquired at 12000, 9000, 6000, 4200, and 3300 ft in the afternoon of the 16<sup>th</sup> of September. The across-track resolutions are 2.0, 1.5, 1.0, 0.7, and 0.5 m, while the along-track resolution is approximately 0.4 m. For the overpasses at 12000 and 6000 ft, the across-track resolution is represented on the x-axis, whereas for the others it is represented on the y-axis. Black circles indicate an artifact caused by a white object


located at the release site. This artifact arises because, first, the reflectance signal appears to correlate with the  $CH_4$  signal, and second, the high albedo of the object leads to increased radiance, which in turn produces an artificially elevated enhancement in the  $CH_4$  maps. At higher altitudes (12000 and 9000 ft), the spatial resolution is too coarse to clearly distinguish this artefact from a true enhancement. The  $CH_4$  plume from the release event with an emission rate of 9.23 kg  $CH_4$  h<sup>-1</sup> is only visible at higher spatial resolutions during overpasses at 3300 and 4200 ft.

Figure 11. RGB images and CH<sub>4</sub> maps for different flight altitudes. All observations are from a release event on the  $16^{th}$  of September with reported emissions of 9.23 kg CH<sub>4</sub> h<sup>-1</sup>

#### 3.5.2 Cast shadows

The cast shadows of buildings and objects are clearly visible at high resolution. Shadows compromise the CH<sub>4</sub> retrieval, which assumes a non-scattering atmosphere, since light in shadowed areas originates solely from scattering. Therefore, an efficient shadow masking is necessary at these resolutions. Figure 12 shows the effect of the shadow mask for a scene with water bodies and cast shadows and the release site with a nearby photovoltaic plant. It can be seen that the masking of cast shadows and dark surfaces such as solar panels is important to prevent biases in the CH<sub>4</sub> maps which would interfere with plume detection. Furthermore, in instances where the plume coincides with shadowed areas, the artificially elevated enhancements would skew emission estimates. As a result of the shadow mask, CH<sub>4</sub> emissions can also be estimated if the plume is transported over shadowed areas.

The downside of shadow masking is that some short plumes of small release events could not be detected because they aligned with shadows. Furthermore, depending on the threshold used for shadow masking, surfaces with low albedos could be masked, preventing the detection of CH<sub>4</sub> emissions.

Figure 12. Examples of scenes containing cast shadows and water bodies (upper row) and the release site during a release event with  $56.7 \text{ kg CH}_4 \text{ h}^{-1}$  (lower row), without and with shadow masking.

#### 3.5.3 Cloud shadows





During all eight overpasses of the  $80.1~\rm kg~CH_4~h^{-1}$  release on the  $19^{th}$  of September, cloud shadows intersected the flight line while on five out of eight overpasses, cumulus clouds obscured the sun over the release site. Under such conditions, the measured radiance is dominated by scattered light, violating the assumptions used in calculating the target spectrum. Moreover, cloud shadows on the flight line render the mean spectrum  $\hat{\mu}$  unrepresentative of the observed radiance  $L_{\rm obs}$  over the release site. Consequently, subtracting  $\hat{\mu}$  from  $L_{\rm obs}$  in Eq. 2 partially removes the CH<sub>4</sub> signal. This effect is evident in Figure 13, which contrasts an overpass with obscured sun at 13:00 UTC with a clear-sun overpass at 12:54 UTC. The lower row shows  $L_{\rm obs} - \hat{\mu}$  over the same plume-free area in both cases. As can be seen, the cloud shadow strongly reduces the signal. As a result, emission estimates for shadowed cases, or for scenes with a substantial fraction of cloud shadows along the flight line, tend to be underestimated. Consequently, a refined retrieval algorithm would be necessary to provide unbiased CH<sub>4</sub> maps and emission estimates.

# 3.5.4 Plume shadows

As a consequence of the unprecedentedly high spatial resolution of AVIRIS-4 and the high SZA for some of the overpasses (see Figure 2), we discovered that, out of 68 detected plumes, 13 were found to contain two plumes that were occasionally overlapping and occasionally distinct, as illustrated in Figure 14. This phenomenon can be explained as plume shadows: One plume appears at the actual release location and corresponds to the CH<sub>4</sub> absorption signal of the light path that first travels from the sun to the ground and, after being reflected, passes through the plume. The second plume is observable at the upper end of the shadow cast by the pole of the source. This plume corresponds to the absorption signal of the light that first passes through the plume, is then reflected from the ground and reaches the sensor without passing through the plume a second time. This

Figure 13. RGB image of 80.1 kg  ${\rm CH_4~h^{-1}}$  release on the  $19^{th}$  of September (a) with and (b) without cloud shadow. The lower row shows the mean  $L_{\rm obs} - \hat{\mu}$  over the same plume-free area within the wavelength window used for  ${\rm CH_4}$  retrieval for both cases.

**Figure 14.** Left: RGB image of the study site with a marker on the release location at 6.5 m above ground. Right: CH<sub>4</sub> map of the study site with two plumes visible.

phenomenon has been shown in simulations by Schwaerzel et al. (2020) and first observed by Sánchez-García et al. (2022). It is important to note that the effect of light passing through the plume only once instead of twice occurs under all conditions with sufficiently high SZA. For sensors with coarse spatial resolution, however, the plume and its shadow cannot be resolved separately and have therefore never been explicitly considered in CH<sub>4</sub> retrieval or emission estimation prior to this study.

#### 3.5.5 MF vs. LMF

For the analysis of this study we also tested the LMF which was developed by Schaum (2021) and tested in Pei et al. (2023).

The plume images using the MF and LMF in Figure 15 show that smaller enhancements (upper row) can be detected more reliably and accurately using the LMF as worked out in Schaum (2021). In our case, the LMF enabled the detection of a release as small as 1.45,kg CH<sub>4</sub> h<sup>-1</sup> at a flight altitude of 4200 ft. This improved detectability can be attributed, in part, to reduced noise levels in the CH<sub>4</sub> retrievals, which facilitated more confident identification of the plume signal. However, the


LMF also increases biases in background CH<sub>4</sub> values compared to the MF as evident in both the upper and lower rows of Figure 15. Additionally, we observed that the LMF had little to no effect on CH<sub>4</sub> enhancements for the largest release events in the campaign, such as the 290 kg CH<sub>4</sub> h<sup>-1</sup> release. This is likely because these stronger enhancements remained within the linear absorption regime of CH<sub>4</sub>, where the standard MF already performs effectively.

**Figure 15.** RGB images and  $CH_4$  maps obtained from MF and LMF for release events with 1.45 and 56.7 kg  $CH_4$  h<sup>-1</sup>, observed at 4200 and 3300 ft. Note that the upper row shows a zoomed-in subsection of the scene to be able to see the short plume.

# 3.6 Estimating the source height from (plume) shadows

The high spatial resolution of AVIRIS-4 offers the unique opportunity to estimate the height h of an emission source based on the length of the shadow  $l_s$  in the RGB image (Figure 14a) cast by the emission source using trigonometry:

$$h = \frac{l_s}{tan(SZA)} \tag{22}$$

Alternatively, the height can be estimated in the same way from the horizontal separation of the starting points of the two plumes (Figure 14b). With increasing distance, the two plumes move together more closely, suggesting that the plume is pushed towards the surface directly after the release. Knowledge of the emission height is an important parameter for emission estimation, as it can be used to determine the effective wind speed, which is a critical input for estimation estimation. In the example shown in Figure 14 with an SZA of  $50^{\circ}$  and a spatial resolution of  $0.53 \,\mathrm{m}$ , the emission plume at the stack must be  $6.4 \pm 1.1 \,\mathrm{m}$  above ground which is in agreement with the true emission height of  $6.5 \,\mathrm{m}$ .

We assume that this technique can be applied if the length of the shadow is at least twice as large as the uncertainty in the length of the shadow  $\sigma_{l_s}$ . The uncertainty in turn is given as the number of pixels with resolution  $l_p$  which contain both shadow and illuminated surface, which is at most one at the top and one at the bottom of the shadow. Therefore, the minimum emission

height becomes

$$h > \frac{2 \cdot l_p}{tan(SZA)} \tag{23}$$

For the campaign discussed in this study, this minimum height is shown in Figure 16.

**Figure 16.** Minimum source height in metres above ground at which shadows of emission sources extend over more than one pixel, shown as a function of SZA and flight altitude above mean sea level.

# 4 Discussion




# 4.1 Capabilities and limitations of AVIRIS-4

Following the success of the AVIRIS Classic and AVIRIS-NG sensors in detecting and quantifying CH<sub>4</sub> emissions as demonstrated in numerous previous studies, this study explores the potential of their successor, AVIRIS-4. Although AVIRIS-4 was primarily developed for surface and vegetation studies, our results show that CH<sub>4</sub> columns can be determined with an unprecedentedly high spatial resolution, enabling the detection of short plumes from low intensity sources. In combination with the enhanced SNR, the detection limit is reduced to 5.5 kg CH<sub>4</sub> h<sup>-1</sup> under good weather conditions and down to below 1.5 kg CH<sub>4</sub> h<sup>-1</sup> under ideal conditions. Because the campaign took place in mid-September, we expect the detection limit could be further reduced under more favourable illumination conditions. As discussed in Section 3.3, we were not able to determine an altitude-dependent detection limit for AVIRIS-4, which complicates direct comparisons with other airborne sensors. For instance, studies with AVIRIS-NG operated at altitudes between 3000 and 6000 m report detection limits of 10-16 kg CH<sub>4</sub> h<sup>-1</sup> under favourable wind conditions (e.g. Ayasse et al., 2023; Conrad et al., 2023). In our case, the lowest release of 9.23 kg CH<sub>4</sub> h<sup>-1</sup> could not be detected at a comparable altitude of 2740 m, likely due to higher wind speeds. This makes it difficult to assess whether and by how much the detection limit has improved. In Thorpe et al. (2016), the lowest detected release was 2.3 kg CH<sub>4</sub> h<sup>-1</sup>, but at a much lower flight altitude of 430 m and under higher wind speeds of 3-5 m s<sup>-1</sup>. Kuhlmann et al. (2025) report a detection limit of 15 kg CH<sub>4</sub> h<sup>-1</sup> at a flight altitude of 6000 m with wind speeds of 0.5 m s<sup>-1</sup>. Overall,






450 comparing detection limits across studies is challenging, as they depend strongly on flight altitude, wind speed, and spectral albedo (Conrad et al., 2023).

Decreasing the detection limit is pivotal because low intensity  $CH_4$  sources are more numerous than high-emitting ones (e.g., Williams et al., 2025; Kuhlmann et al., 2025). Consequently, accurate estimates of total  $CH_4$  emissions depend on detecting smaller sources. For instance, based on the best detection limit of AVIRIS-NG of 15 kg  $CH_4$  h<sup>-1</sup> reported in Kuhlmann et al. (2025) and the distribution of oil production sites in Romania across the outlined scenarios, AVIRIS-NG was able to detect between 45% and 62% of total emissions. In contrast, assuming a detection limit of 5.5 kg  $CH_4$  h<sup>-1</sup>, AVIRIS-4 would increase this detection coverage to approximately 67%–81%. Moreover, the detection limit of 5.5 kg  $CH_4$  h<sup>-1</sup> achieved by AVIRIS-4 effectively enables the identification of all point sources listed in the E-PRTR registry, which mandates reporting for emissions exceeding 100000 kg  $CH_4$  yr<sup>-1</sup> (equivalent to 11.4 kg  $CH_4$  h<sup>-1</sup>) (European Parliament and the Council of the European Union).

In comparison with its predecessor, AVIRIS-4 has compromised some of its spectral resolution in order to enhance its spatial resolution and SNR. This study, along with comparisons to other airborne imaging spectrometers with higher spectral but lower spatial resolution such as MAMAP2DL (e.g. Krautwurst et al., 2024), demonstrates that this trade-off is beneficial for detecting small-scale CH<sub>4</sub> enhancements from low intensity sources, whose plumes typically extend only a few decimetres to a few metres.

The noise level of the CH<sub>4</sub> maps was estimated as the standard deviation of the retrieved columns over the brightest 50% of pixels. This resulted in a noise level of AVIRIS-4 of 700 ppm-m at an average resolution of 0.5 m which is comparable to reported values of AVIRIS-NG at 5 m resolution for suboptimal illumination conditions (e.g. Borchardt et al., 2021). Such noise levels are expected, given that the campaign was conducted in mid-September under low solar zenith angles (SZAs). In addition, negative values were not masked during the CH<sub>4</sub> retrieval, which increases the apparent noise.

The current study also shows that owing to the higher SNR and higher spatial resolution, emissions can also be detected and estimated with less illumination and under suboptimal surface and atmospheric conditions, which are characterised by inhomogeneous albedo, strong turbulence, cast shadows and cloud shadows (see Section 3.1), compared to previous controlled release experiments with AVIRIS-NG (e.g. Thorpe et al., 2016; Duren et al., 2019). For example, the higher spatial resolution allows for a more accurate filtering for shadow pixels and albedo artefacts which, if undetected, could lead to biases in emission estimates, as outlined in Section 3.5.2. This capability allows AVIRIS-4 to be effectively applied to built-up sites with heterogeneous surface albedo and cast shadows, conditions commonly encountered around CH<sub>4</sub> sources in the oil, gas, and coal mining sectors. However, the higher spatial resolution also results in new challenges. One of them is the double plume originating from plume shadows illustrated in Figure 14. We corrected for this artifact when the true plume and its shadow were clearly separated, but its impact on retrieved CH<sub>4</sub> enhancements requires further analysis. Additional work is needed to handle partially overlapping plumes, especially as the phenomenon of plume shadows also affects instruments with coarser spatial resolution even if they do not spatially resolve the plume shadow (Schwaerzel et al., 2020). A second challenge that arises with higher spatial resolution are the higher per-pixel enhancements for larger sources. As a result, the linearisation of the unit absorption spectrum no longer holds and assumed enhancements for the calculation of the absorption spectrum have greater





influences on the retrieved enhancements. Therefore, careful selection of the assumed enhancements, e.g. with the iterative approach used in this study, is essential.

Lastly, the current study shows mixed results when using the LMF introduced by Schaum (2021). On the one hand, the proposed improvement for the detection of weak plumes was also observed in this study and lowered the detection limit even under challenging conditions. On the other hand, the LMF increased local biases in the retrieved CH<sub>4</sub> maps which would impede the (automated) detection of plumes as it introduces elevated background levels that can obscure smaller enhancements or cause false positives in regions with complex spatial patterns. Additionally, the improved performance of the LMF for larger sources reported by Schaum (2021) was not observed. Further systematic analyses will be required to determine under which circumstances the LMF method can outperform the more established MF.

# 4.2 Wind speed estimation

As seen in Section 3.4, estimated emissions linearly depend on the wind speeds used. Therefore, accurate estimates of wind speeds are crucial for accurate emission estimates. Additionally, our analysis demonstrated that uncertainties in wind speeds contributed disproportionately to the uncertainty of the estimated emissions. Based on the analysis of this study, the wind speed representation error ( $\sigma_{repr}$ ), uncertainties in effective wind speed ( $\sigma_{eff}$ ) and instrument precision ( $\sigma_{inst}$ ) likely need to be revised upward. Consequently, building on the understanding of wind speed inputs (see Section 4.2.1), future research on emission estimation from remote sensing data should prioritise methods for deriving the effective wind speed that governs plume transport (see Section 4.2.2).

#### 4.2.1 Source of wind speed estimates

As clearly illustrated in Figure A1, near-surface winds can be be highly variable and gusty. We frequently found that  $U_{10}$  measured by the Lidar varied between 1.0 and  $3.0\,m\,s^{-1}$  within one minute. These rapid fluctuations highlight that reanalysis wind fields are insufficient for high-resolution emission estimates with new-generation sensors, as they can introduce substantial biases. A high-resolution model may be able to represent this gustiness more realistically in a statistical sense, but capturing the actual wind conditions at the moment of the overpass remains practically impossible. Alternatively, wind speed data from existing measurement networks could be used for emission estimation. However, these networks have varying data quality and might not be available in the vicinity of a CH<sub>4</sub> source. Therefore, one could employ mobile instruments as it was used for the controlled release experiment in the current study. Even if this would provide the most accurate estimate of the wind speed, setting up wind speed instrument would negate the advantage of remote sensing instruments which is to image extensive areas and estimate the emissions of a large number of sources. Additionally, the current analysis has shown that under turbulent conditions, wind speed representation errors can be substantial, even when wind measurements are taken just 100 m from the source. Therefore, the best approach would be to measure wind speed profiles in tandem with imaging spectrometry, e.g. by using an airborne wind lidar as investigated in Thorpe et al. (2021).






# 4.2.2 Effective wind speeds

In addition to determining the small-scale and short-term wind speeds, a further challenge is to determine the effective wind speed at which the plume was transported. Although an increasing number of studies attempt to derive Ueff from model simulations (e.g. Varon et al., 2018; Guanter et al., 2021; Sánchez-García et al., 2022; Ayasse et al., 2023; Guanter et al., 2025), none has systematically investigated the effect of emission height, atmospheric stability or surface roughness on Ueff. Moreover, existing simulations lack the spatial and temporal resolution required for AVIRIS-4 applications. To advance our understanding of the effective wind speed, high-resolution model studies are needed to analyse the impact of the aforementioned factors. Ideally, these results could be parametrised to estimate the effective wind speed based on known driving factors. While estimates for the 3D wind field, surface roughness and heat fluxes could be obtained from regional weather prediction models, information about the emission height could be obtained directly from AVIRIS-4 imagery as outlined in Section 3.6. Another innovative approach has recently been demonstrated in Eastwood et al. (2025) with AVIRIS-3 where a single plume was observed multiple times during one overpass by adjusting the flight path of the aircraft. Specifically, the aircraft ascended while approaching the plume, maintained a level trajectory while flying directly over it, and then descended after passing it. From the resulting three images, the plume velocity was estimated by calculating optical flow vectors for consecutive CH<sub>4</sub> images. While this method proved to significantly improve the estimates of the effective wind speed compared to reanalysis data and on-site wind lidar data, it requires a-priori knowledge of the source location to plan the required flight manoeuvres. One workaround would be to use real-time in-flight retrieval of CH<sub>4</sub> (e.g. Thompson et al., 2015) in combination with pitching AVIRIS-4 using the already installed stabilisation platform.

# 5 Conclusions

Detecting and quantifying the emissions from a large number of sources is essential for obtaining accurate inventories of  $CH_4$  emissions. The current study shows that AVIRIS-4 can be used for the improved detection of  $CH_4$  emissions and subsequent quantification. The combination of high spatial resolution with the unprecedentedly high SNR of AVIRIS-4 decreases the detection limit of AVIRIS-4 to below 5.5 kg  $CH_4$  h<sup>-1</sup> under good weather conditions and down to 1.5 kg  $CH_4$  h<sup>-1</sup> under ideal conditions which effectively enables the identification of all point sources listed in the E-PRTR registry. As a result, previously undetected low intensity and dispersed sources can be identified and accounted for in emission budgets. We demonstrate that the high spatial resolution of AVIRIS-4 enables its effective use under challenging conditions and in heterogeneous environments, which are frequently encountered in real-world applications. Furthermore, we show how high-resolution AVIRIS-4 data can be used for the estimation of the source height which is critical information when estimating the effective wind speed. As with earlier sensors and algorithms, emission estimation with AVIRIS-4 is affected by uncertainties in the estimation of the effective wind speed - especially at the short length and timescales presented in this study. Overall, this study highlights that AVIRIS-4 represents a significant step forward in airborne methane remote sensing, offering unprecedented sensitivity to low-intensity sources under challenging conditions. At the same time, it underscores the importance of advancing wind speed estimation techniques and improving retrieval strategies to fully exploit the sensor's capabilities. Future work should therefore

focus on integrating AVIRIS-4 observations with dedicated wind measurements and adapting the  $CH_4$  retrieval algorithm to the unprecedentedly high spatial resolution.

Code availability. The ddeq version 1.0 used for this study is available on Gitlab.com (https://gitlab.com/empa503/remote-sensing/ddeq). The code for AVIRIS-4 data processing and CH<sub>4</sub> retrieval is available on request.

Data availability. ERA5 data are available at https://doi.org/10.24381/cds.adbb2d47 (Hersbach et al., 2018). The retrieved AVIRIS-4 CH<sub>4</sub> maps, wind and sources data and estimated emissions are available on the Zenodo data repository (DOI: 10.5281/zenodo.16410532).

Figure A1 reveals substantial systematic deviations, particularly during daytime, likely caused by small- to mesoscale atmospheric circulations influenced by local terrain. Such features are not captured by the relatively coarse spatial  $(0.25^{\circ} \times 0.25^{\circ})$  and temporal resolution of ERA5. Furthermore, ERA5 fails to resolve turbulent fluctuations in near-surface winds that are evident in lidar observations.

Figure A1. ERA5  $U_{10}$  vs. on-site lidar  $U_{10}$ . The blue shaded area represents the ERA5 ensemble spread while the red shaded area depicts the min and max wind speed for 1 minute intervals.

Figure A2. Top row: Relative contribution of the individual uncertainty terms of the CSF and IME to the uncertainty of the estimated emissions Q. Bottom row: wind speed uncertainty contributions by natural variability  $\sigma_{\text{var}}$ , effective wind speed  $\sigma_{\text{eff}}$  and instrument precision  $\sigma_{\text{inst}}$ 

**Figure A3.** Pair plot of the wind speeds measured by the wind lidar at 10 and 20 m as well as from a meteorological station affixed to the lidar, approximately 1 m off the ground.

Lidar 10 m wind speed [m  $s^{-1}$ ]

|                    |       | 16.09.2024    | 17.09.2024    | 18.09.2024    | 19.09.2024    | 20.09.2024 |
|--------------------|-------|---------------|---------------|---------------|---------------|------------|
| Local time [UTC+2] | 09:00 | N/A           | 1.1 ± 0.5     | $1.7 \pm 0.6$ | 1.5 ± 0.3     | 1.8 ± 0.4  |
|                    | 10:00 | N/A           | $1.4 \pm 0.4$ | $0.9 \pm 0.3$ | $0.9 \pm 0.3$ | 1.6 ± 0.4  |
|                    | 11:00 | 1.7 ± 0.7     | 0.9 ± 0.3     | 0.9 ± 0.3     | $0.8 \pm 0.3$ | 0.8 ± 0.3  |
|                    | 12:00 | 2.1 ± 0.8     | 1.1 ± 0.5     | 1.4 ± 0.5     | 1.1 ± 0.4     | 0.8 ± 0.4  |
|                    | 13:00 | 2.5 ± 0.9     | 2.2 ± 0.8     | 1.5 ± 0.6     | $1.2 \pm 0.6$ | 1.0 ± 0.5  |
|                    | 14:00 | $2.3 \pm 0.9$ | 1.5 ± 0.6     | 1.7 ± 0.7     | $2.0 \pm 0.8$ | 2.2 ± 0.6  |
|                    | 15:00 | $3.2 \pm 0.9$ | 1.3 ± 0.5     | $2.0 \pm 0.9$ | $2.1 \pm 1.0$ | 1.8 ± 0.6  |
|                    | 16:00 | 2.9 ± 0.8     | 1.2 ± 0.5     | 2.0 ± 0.8     | 2.4 ± 1.0     | 1.5 ± 0.6  |
|                    | 17:00 | N/A           | N/A           | 2.1 ± 0.9     | 3.1 ± 0.9     | N/A        |

**Figure A4.** Average and standard deviation of lidar 10 m wind speed during each release in local time [UTC+2]. Wind data has been resampled to 1 minute intervals.

Lidar 10 m wind direction [°]

|                    |       | 16.09.2024 | 17.09.2024 | 18.09.2024 | 19.09.2024 | 20.09.2024 |
|--------------------|-------|------------|------------|------------|------------|------------|
| Local time [UTC+2] | 09:00 | N/A        | 15 ± 40    | 112 ± 26   | 153 ± 16   | 135 ± 78   |
|                    | 10:00 | N/A        | 303 ± 32   | 143 ± 56   | 93 ± 44    | 131 ± 24   |
|                    | 11:00 | 113 ± 35   | 292 ± 68   | 213 ± 52   | 122 ± 52   | 169 ± 61   |
|                    | 12:00 | 139 ± 48   | 318 ± 63   | 246 ± 47   | 172 ± 45   | 222 ± 50   |
|                    | 13:00 | 315 ± 38   | 321 ± 42   | 187 ± 45   | 236 ± 64   | 317 ± 55   |
|                    | 14:00 | 319 ± 37   | 330 ± 44   | 130 ± 76   | 55 ± 39    | 301 ± 12   |
|                    | 15:00 | 317 ± 24   | 258 ± 43   | 86 ± 34    | 57 ± 46    | 336 ± 44   |
|                    | 16:00 | 329 ± 26   | 226 ± 39   | 66 ± 53    | 72 ± 33    | 114 ± 43   |
|                    | 17:00 | N/A        | N/A        | 53 ± 34    | 73 ± 28    | N/A        |

**Figure A5.** Average and standard deviation of lidar 10 m wind direction during each release in local time [UTC+2]. Wind data has been resampled to 1 minute intervals.

Figure A6. Wind roses of lidar 10 m wind speed and direction during each release in local time [UTC+2].

Figure A7. Uncertainty weighted average estimates for each release using  $U_{\text{eff}}$  derived in this paper. The number of observations n, emission height h, plume length l and average wind speed u are indicated above each bar.

Author contributions. SM conducted the analysis and wrote the paper with input from all co-authors; MV and AH planned and organised the
 flight campaign and processed the data to georeferenced level 1 data; AM, AB, CJ and VB organised and conducted the controlled release experiment; DB was involved in the planning of the campaign; GK coordinated and supervised the project.

Competing interests. At least one of the (co-)authors is a member of the editorial board of Atmospheric Measurement Techniques.

Acknowledgements. This research has been funded under the framework of UNEP's International Methane Emissions Observatory (IMEO) (agreement no. CCD24-MB7279). Data processing and storage were performed at the Swiss National Supercomputing Centre (CSCS). The

AI tools ChatGPT and DeepL were used for grammar checking, copy-editing, and minor wording improvements.

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
