# Peer review of "Quantifying CH4 point source emissions with airborne remote sensing: First results from AVIRIS-4"

_EGUsphere, 2025_

## Author Comment (AC1)

**Answer to the reviewers:**

**Reviewer 1**

AC: We like to thank Reviewer 1 for the valuable review and the detailed comments. We have revised the manuscript accordingly and provide a point-by-point response to all comments below. Our replies are written in blue.

**Specific Comments**

**1. On Retrieval Algorithm Limitations**

**(1) Linearity Assumption:** The matched filter (MF) and lognormal matched filter (LMF) are based on a linearized Beer-Lambert law, which may be invalid for the strong, localized enhancements from intense sources resolved by AVIRIS-4. The iterative approach refines the input spectrum but does not overcome this fundamental theoretical constraint. The authors should assess the severity of potential non-linear absorption effects for the strongest plumes (e.g., 290 kg $CH_4$ h$^{-1}$) to better define the operational limits of their retrievals.

The motivation for this study was our participation in the release experiment, which required providing results a few weeks after the flights. Consequently, our method was adapted from our established AVIRIS-NG processing chain (Kuhlmann et al. 2025). While we have implemented several important modifications to account for the higher spatial resolution of AVIRIS-4 (such as cast shadow masking, iterative matched filter, plume shadow correction), it was beyond the scope of this study to address all methodological challenges associated with the new sensor. Instead, we provide a list of discussion points for future developments and are happy to include the points raised by the reviewer to this list.

As the reviewer correctly pointed out, the MF is based on the linearisation of the Beer-Lambert law. Conversely, the LMF is not. The linearisation error for the MF can be computed as follows:

$$L_{exact}(\lambda) = L_0(\lambda) \cdot e^{-\tau(\lambda)}$$

Here, we use $\tau(\lambda) = \alpha \cdot s(\lambda)$, assuming that the optical depth linearly depends on the $CH_4$ enhancement. This assumption is justified when using a high-resolution input absorption spectrum $s(\lambda)$, for which individual absorption features are well resolved and saturation effects are negligible. This condition is satisfied for the spectrum employed in this study, which has a spectral resolution of 0.001 nm.

When linearising at $\alpha = 0$, we can approximate $L(\lambda)$:

$$L_{linear}(\lambda) = L_0(\lambda) \cdot (1 - \tau(\lambda))$$

The linearisation error is then calculated as

$$\delta L(\lambda) = L_{exact} - L_{linear}(\lambda) = L_0(\lambda) \cdot e^{-\tau(\lambda)} - L_0(\lambda) \cdot \left(1 - \tau(\lambda)\right) = L_0(\lambda) \cdot \left(e^{-\tau(\lambda)} - (1 - \tau)\right)$$

Using the remainder of the Taylor series expansion, $\delta L(\lambda)$ can be approximated by

$$\delta L(\lambda) \approx L_0(\lambda) \cdot \frac{\tau^2}{2}$$

However, when using our iterative approach to determine $s(\lambda)$, we no longer linearise at $\alpha = 0$ but at $\alpha_i$:

$$L(\lambda) = L_0(\lambda) \cdot e^{-\alpha_i s}$$

$$L_{\alpha_i}(\lambda) \approx L(\lambda) + \frac{dL}{ds}\bigg|_{\alpha_i} (\alpha - \alpha_i)$$

$$\frac{dL}{ds}\bigg|_{\alpha_i} = -L_0(\lambda) \cdot \alpha_i \cdot e^{-\alpha_i s}$$

$$L_{\alpha_i}(\lambda) \approx L(\lambda) - L_0(\lambda) \cdot \alpha_i \cdot e^{-\alpha_i s}(\alpha - \alpha_i)$$

Therefore, the linearisation error at $\alpha_i$ becomes

$$\delta L(\lambda) = L(\lambda) - L_{\alpha_i}(\lambda) = L_0(\lambda) \cdot e^{-\alpha s} - \left(L_0(\lambda) \cdot e^{-\alpha_i s} - L_0(\lambda) \cdot \alpha_i \cdot e^{-\alpha_i s}(\alpha - \alpha_i)\right)$$

This can be approximated using Taylor expansion as

$$\delta L(\lambda) \approx \frac{1}{2} \cdot L_0(\lambda) \cdot s_0^2 \cdot e^{-\alpha_i s}(\alpha - \alpha_i)^2$$

If we choose an $\alpha_i$ close to the true $\alpha$, the error $\delta L(\lambda)$ goes towards 0.

To verify this, we generated a synthetic radiance dataset and embedded a $CH_4$ plume using the absorption spectrum of $CH_4$. Then, we applied the MF and LMF to this data. Figure 1 (a) shows the retrieved mean $CH_4$ enhancements within the plume using the MF and LMF. It can be seen that both the MF and LMF yield comparable results but the MF tends to slightly underestimate the emissions for higher enhancements. This is not only the result of the remainder of the linearisation error but a contamination of the mean spectrum for plumes with higher enhancements. This is shown in Figure 1 (b), which presents estimated $CH_4$ enhancements for a case with true enhancements of 20 ppm as a function of the fraction of the scene containing a $CH_4$ plume. For larger fractions, more of the $CH_4$ absorption signal "leaks" into the mean spectrum and leads to an underestimation of the $CH_4$ enhancements. Conversely, due to the log transformation of the radiance data, the LMF is much more robust to this contamination. In the case of our study, the contamination likely does not have a significant influence on the retrieved $CH_4$ enhancements as the mean is calculated over >23000 pixels in along-track direction whereas the plumes only consist of a few dozen pixels, i.e. <0.05% of the scene. Additionally, the strongest release only had mean $CH_4$ enhancements below 20 ppm.

[Figure]

**Figure 1: Comparison of CH₄ enhancements from MF and LMF for synthetic data. (a) Estimated CH₄ enhancement using MF and LMF. (b) Dependence of estimated CH₄ enhancement for increasing fractions of the scene containing a plume with a true CH₄ enhancement of 20ppm.**

Based on the findings above, we have added a short explanation of this in the Methods section to clarify the validity range of our iterative approach:

"Our iterative approach reduces the approximation error introduced by the linearisation of the Beer-Lambert law by expanding around the current estimate of $\alpha$ rather than $\alpha = 0$, which decreases the linearisation error quadratically in the update step. For large enhancements, this substantially mitigates non-linear absorption effects."

We have also specified that we use the LMF only exploratively as a comprehensive analysis is beyond the scope of this paper:

"In the present study, we evaluated the LMF only exploratively to illustrate its behaviour on AVIRIS-4 data with an emphasis on the smallest and largest release events."

Lastly, we added the derivation and Figure 1 to the supplement.

**(2) LMF Background Bias:** The observed background bias introduced by the LMF is a significant concern, likely stemming from the log-transform amplifying noise in low-SNR pixels. This is a fundamental signal-processing issue, not merely a challenge for automation. A comparative analysis of the noise covariance matrix ($\hat{S}$) in linear versus log space for different surface albedos is needed to elucidate the mechanistic origin of this bias.

To address the reviewer's concern regarding the origin of the LMF-induced background bias, we analysed how the log transform alters noise propagation in the LMF and the resulting background covariance structure compared to the MF:

In linear space, a radiance measurement $x$ can be separated into the mean radiance $\mu$, the target spectrum $s$ and the noise $\epsilon$:

$$x = \mu + s + \epsilon$$

However, in log space, the measurement is transformed to

$$y = \log(x) = \log(\mu + s + \epsilon) \approx \log(\mu + s) + \frac{\epsilon}{\mu + s} + O\left(\frac{\epsilon^2}{(\mu + s)^2}\right)$$

If $s \ll \mu$, we obtain

$$y \approx \log(\mu) + \frac{s}{\mu} + \frac{\epsilon}{\mu}$$

As we see, the noise is scaled by $\frac{1}{\mu}$ which means that as the mean radiance $\mu$ approaches zero, the effective noise term approaches infinity.

Because the MF and LMF detection statistics depend explicitly on the inverse of the background covariance matrix $\hat{S}^{-1}$, understanding the bias introduced by the log transform requires analysing how noise variance is distributed across spectral dimensions. Therefore, we performed a comparative eigenanalysis of the background covariance matrix for four albedo classes in the $CH_4$ absorption window. They are characterised by their mean radiance as low (4.95e-05 $W/m^2/sr/nm$), mid (2.43e-03 $W/m^2/sr/nm$), high (6.98e-03 $W/m^2/sr/nm$) and very high: (1.07e-02 $W/m^2/sr/nm$). The scree plots in Figure 2 show that for dark surfaces, the noise covariance eigenvalues increase by several orders of magnitude when moving from linear to log space and their decay becomes significantly flatter. This demonstrates that the log transform amplifies noise dramatically for pixels with low SNR, distributing that noise across many principal components.

[Figure]

**Figure 2: Scree plots for MF and LMF for different mean SWIR radiance levels between 1480-1800 and 2080-2500nm.**

Based on these insights, we have adjusted the respective section in the results to elucidate the likely origin of this higher background bias:

"An analysis of the eigenvalues of the covariance matrices for different surface albedos revealed that these biases likely stem from the fact that the log transform amplifies noise dramatically for pixels with low SNR."

We also adjusted the following section in the discussion:

"On the other hand, the LMF increased local biases in the retrieved $CH_4$ maps which we attribute to the amplification of noise by the log-transform in pixels with low SNR, caused by low albedo. This spatially more heterogeneous background can obscure small enhancements or produce false detections. In contrast to Schaum (2021), we did not observe an improved performance of the LMF for large release events. The iterative MF applied in our study seems to successfully account for most non-linear absorption in pixels with large $CH_4$ enhancement."

**2. On Radiative Transfer at High Spatial Resolution**

**(1) Plume Shadow & Surface Heterogeneity:** The plume shadow is a noteworthy finding. However, the geometric correction (Eqs. 6-7) assumes a uniform surface albedo. In real-world scenarios, the plume and its shadow often overlie heterogeneous surfaces (e.g., vegetation vs. asphalt), meaning the two light paths experience different ground reflectances. This will introduce errors. The authors should discuss this limitation and its potential impact on quantification accuracy.

Our geometric correction assumes a non-scattering atmosphere, which implies no albedo dependency. This is consistent with our $CH_4$ retrieval, which also assumes a non-scattering atmosphere.

The assumption of a non-scattering atmosphere is sufficient for the wavelengths used for the $CH_4$ retrieval, as atmospheric scattering is small at these wavelengths. The effect of scattering is less than 5% even for high aerosol concentrations (compare Figure 2 in Kuhlmann et al., 2025).

**(2) 3D Effects:** At AVIRIS-4's sub-meter resolution, 3D radiative transfer effects, such as adjacency contamination from scattered light, may become non-negligible. The albedo artifact mentioned in Section 3.5.1 indicates such effects. The authors should comment on whether adjacency effects could influence retrieved methane enhancements, particularly within the core of strong, compact plumes.

We agree with the reviewer that at the high resolution of AVIRIS-4, adjacency effects due to scattering become increasingly important for small wavelengths. However, as mentioned above, the effect of scattering can be neglected at the wavelengths used for the $CH_4$ retrieval (see Figure 2 in Kuhlmann et al., 2025). Therefore, our retrieval assumes a non-scattering atmosphere, which would not be affected by adjacency effects.

The shown artifact in section 3.5.1. is likely not caused by adjacency effects as it is visible for all flight altitudes. It is more likely that the artifact is caused by the object with a spectral signature that is closely related to the one of $CH_4$ absorption. Additionally, the object is very bright, which leads to a larger signature which is likely not fully removed by the albedo correction.

**3. Methods and Data Processing**

**(1) MF/LMF Comparison:** The conclusion regarding LMF's performance should be supported by a systematic, quantitative comparison against the standard MF. Statistical metrics (e.g., RMSE, bias distribution across the entire dataset) are needed, rather than reliance on selective examples.

We agree with the reviewer that a full, systematic quantitative comparison between MF and LMF would indeed be necessary for a comprehensive performance evaluation.

However, we would like to clarify that the goal of this paper is not to benchmark LMF against MF, but rather to demonstrate the first use of AVIRIS-4 for $CH_4$ detection and quantification, with LMF included only as an exploratory, secondary experiment to explore if we can see more small sources or if there is a bias in large sources. To avoid implying a full performance assessment, we have revised the respective paragraphs (see answer to question 1).

**(2) Convergence Criteria:** The convergence criteria for the iterative absorption spectrum calculation should be explicitly defined. Please specify the quantitative threshold (e.g., change in mean plume enhancement between iterations) used, rather than stating it converged in "2-3 iterations."

We use the criterion that the iterative process is continued only as long as the relative increase in the computed mean plume enhancements between successive is smaller than 5%, which happens after 2-3 iterations (see Figure 3 below for an example).

We have adjusted the respective section to clarify this threshold:

"The mean enhancement in the detected plume was then used for the subsequent iteration of the matched filter, which converged after three iterations with changes between successive iterations falling below a 5% threshold."

[Figure]

**Figure 3: Convergence of retrieved CH₄ enhancements for the iterative MF.**

**(3) Retrieval Justification:** A brief justification for relying solely on matched filter techniques would strengthen the manuscript. Please comment on why more robust methods like WFM-DOAS were not considered, especially given the potential for non-linear effects from strong sources.

The reason for using the MF was its extraordinary computational efficiency given the large amount of data. Our data consisted of 101 datasets with 1241 across-track pixels and, on average, 23000 along-track pixels. The MF was essential to process the data within a few weeks. WFM-DOAS is an interesting approach that we might consider in future. We have added a sentence in the manuscript on potentially using WFM-DOAS:

"Other approaches, such as WFM-DOAS (e.g. Borchardt et al., 2021), may also help better account for non-linear effects arising from strong emission sources. However, they are computationally more expensive than the MF and tend to work better for sensors with higher spectral resolution."

As mentioned in the answer to question 1, we tried to account for the non-linear behaviour of trace gas absorption using the iterative estimation of the enhancement used for the calculation of the unit absorption spectrum.

**4. Results and Discussion**

**(1) Wind Speed Comparison:** To complement Figures 7 and 8, a summary table with performance metrics (RMSE, MBE) for each wind speed method, stratified by emission rate bins, would provide a clearer and more systematic comparison.

Thank you for the suggestion. We have added a table with the respective values in the supplement.

**(2) Quantifying Plume Shadow Impact:** The discussion on plume shadows would be strengthened by a quantitative analysis. Providing specific values for the emission rate bias (comparing corrected vs. uncorrected estimates for affected plumes) is recommended.

Thank you for pointing that out! We observed clearly separated double plumes in only four overpasses, which are listed below. However, the impact of the correction is (likely) masked by the high variability of the wind speed.

| File name | Correction factor |
|---|---|
| "M017_240919_FRA_Pau_Methane_6000ft_Line_10001_155818_000_rdn" | 2.56 |
| "M017_240919_FRA_Pau_Methane_4200ft_Line_20001_160932_000_rdn" | 2.61 |
| "M017_240919_FRA_Pau_Methane_4200ft_Line_10001_161600_000_rdn" | 2.64 |
| "M017_240919_FRA_Pau_Methane_3000ft_Line_10001_162717_000_rdn" | 2.70 |

We have added the mean correction factor to the revised manuscript:

"To correct for plume shadows, we applied the method outlined in Section 2.4.4 to the four observed plumes that were clearly separated. This resulted in a mean correction factor of 2.6."

**(3) Resolution Trade-off:** The trade-off made by AVIRIS-4 (higher spatial resolution at the cost of spectral resolution) warrants brief discussion. What are the implications for retrieving other gases (e.g., $CO_2$) or for applications over complex surfaces?

In our analysis we have seen many plumes with lengths <2 m which would not have been detected at a coarser spatial resolution due to the limited number of enhanced pixels. Furthermore, the high spatial resolution facilitated the detection of plumes over heterogeneous backgrounds as they could be distinguished from albedo artefacts. Examples can be seen in Figure 11 in the manuscript or in the following images in the supplementary material:

- M014_240916_FRA_Pau_Methane_3000ft_Line_0001_163819_000_rdn
- M014_240916_FRA_Pau_Methane_4200ft_Line_0001_130856_000_rdn
- M014_240916_FRA_Pau_Methane_4200ft_Line_0001_162509_000_rdn

We also tested AVIRIS-4 for the detection of $CO_2$ in an explorative approach. As the campaign was not designed to overpass large sources of $CO_2$ emissions, we had to restrict our test case to one chimney close to the site of the controlled release experiment. There we found that a $CO_2$ plume could be detected using the same MF approach. However, the plume appeared to be noisy which is why binning would have to be applied. However, further research would be needed to determine the effect of the coarser spectral resolution of AVIRIS-4 on the detection of trace gases other than $CH_4$, which is currently out of the scope of this study.

**(4) LMF for Strong Sources:** The finding that LMF offers no improvement for strong sources seems to contradict Schaum (2021), who posits it as the uniformly most powerful detector. This discrepancy should be discussed.

We assume that we don't see an improvement of the LMF compared to the MF for the strongest release events because our iterative MF already compensates for most of the non-linear absorption associated with high optical depths. We have added a sentence in section 3.5.5 clarifying this:

"Additionally, we observed that the LMF had little to no effect on $CH_4$ enhancements for the largest release events in the campaign, such as the 290 kg $CH_4$ h$^{-1}$ release. This is likely because our iterative MF already compensates for most of the non-linear absorption associated with high optical depths."

Additionally, the log transform amplifies the noise for low signal pixels, which reduces its applicability:

"An analysis of the eigenvalues of the covariance matrices for different surface albedos revealed that these biases likely stem from the fact that the log transform amplifies noise dramatically for pixels with low SNR."

**5. Uncertainty Analysis**

**(1) Error Typology:** The decomposition of wind speed uncertainty should more clearly distinguish between systematic (e.g., $\sigma_{eff}$, $\sigma_{inst}$) and random ($\sigma_{var}$) error components, as their impacts on the final emission estimate differ.

This is indeed an important distinction to make. Therefore, we demonstrate how the uncertainties $\sigma_{eff}$, $\sigma_{inst}$ and $\sigma_{var}$ contribute to the overall wind speed uncertainty in Figure A2. We added a brief sentence to the figure for clarification:

"Figure A2 shows that the uncertainty in the wind speed $\sigma_U$ contributes 99.4% to the total uncertainty of the estimated emissions $\sigma_q$ for the CSF and 91.3% for the IME. $\sigma_U$ in turn consists 90.4% of natural wind speed variability $\sigma_{var}$."

**(2) Pixel Area Uncertainty:** The 5% uncertainty estimate for pixel area, based on visual comparison, appears subjective. A more objective quantification, potentially from an analysis of geolocation residuals using ground control points, is recommended.

We added a term for the uncertainty for the pixel area due to a hardware issue of the inertial navigation system (INS) which impeded the automatic matching of AVIRIS-4 data with the INS data. Therefore, this matching had to be done manually to be able to project AVIRIS-4 data onto the digital elevation model (DEM).

To examine the precision of pixel areas, we manually computed the area of a built-up site close to the release site from Google Earth and AVIRIS-4 data. We found a bias of 0.16% and a standard deviation of 0.6%.

However, an estimate of the pixel area precision is not trivial as we did not have ground control points.

For the current study, we used a conservative estimate of 5% to make sure that we would be within the true uncertainty of the pixel area, knowing that the uncertainty in emission estimates due to the pixel area is orders of magnitude smaller compared to the uncertainty in wind speeds (see e.g. Figure A2). To validate this assumption, we recomputed the estimated $CH_4$ emissions after imposing a pixel-area uncertainty of 0.5%. The resulting emission estimates and their associated precision remained unchanged, with differences below ±0.00%, confirming that this source of uncertainty is negligible for the present analysis.

[Figure]

**Figure 4: Comparison of pixel area with Google Earth imagery.**

**6. Minor Points**

**Detection Limit Context:** The abstract and/or conclusions should present a clearer, more direct statement comparing the detection limit of AVIRIS-4 with its predecessor, AVIRIS-NG, as the current phrasing is somewhat vague.

We agree that this comparison was kept quite vague. Therefore, we have added a few statements to the abstract and conclusion highlighting the comparison to AVIRIS-NG:

"This is below the $10 - 16$ kg $CH_4$ $h^{-1}$ detection limits reported for its predecessor AVIRIS-NG in previous studies. In practice, AVIRIS-4 therefore extends the range of reliably detectable point sources by approximately a factor of two to three relative to AVIRIS-NG when flown at low altitudes, which effectively enables the identification of all point sources listed in the E-PRTR registry. As a result, previously undetected low intensity and dispersed sources can be identified and accounted for in emission budgets."

**References:**

Borchardt, J., Gerilowski, K., Krautwurst, S., Bovensmann, H., Thorpe, A. K., Thompson, D. R., Frankenberg, C., Miller, C. E., Duren, R. M., and Burrows, J. P.: Detection and quantification of $CH_4$ plumes using the WFM-DOAS retrieval on AVIRIS-NG hyperspectral data, Atmos. Meas. Tech., 14, 1267–1291, https://doi.org/10.5194/amt-14-1267-2021, 2021.
Kuhlmann, G., Stavropoulou, F., Schwietzke, S., Zavala-Araiza, D., Thorpe, A., Hueni, A., Emmenegger, L., Calcan, A., Röckmann, T., and Brunner, D.: Evidence of successful methane mitigation in one of Europe's most important oil production region, Atmos. Chem. Phys., 25, 5371–5385, https://doi.org/10.5194/acp-25-5371-2025, 2025.

**Reviewer 2**

AC: We would like to thank Reviewer 2 for their helpful comments. We have revised the manuscript based on their suggestions and provide a point-by-point response below. Our replies are written in blue.

The manuscript by Meier et al. deals with a methane controlled-released experiment using the AVIRIS-4 airborne spectrometer. The authors report on the performance of AVIRIS-4 for the detection and quantification of methane plumes, and also use the generated dataset of plume detections for the illustration and evaluation of several technical aspects related to the remote sensing of methane point sources.

Overall I think it is a nice study with several important messages for the growing methane remote sensing community. The manuscript is well written and presented, and the topic fits perfectly in AMT, so I recommend its publication.

I would like to request the authors to address the following points in their revision of the manuscript:

1. Plume shadows (section 2.4.4 and 3.5.4): the authors illustrate the issue of "plume shadows" in their dataset. However, the discussion of this effect and the implications for plume detection, attribution and quantification remain quite superficial. I would recommend the authors to deepen the discussion of this effect. In particular, the authors could discuss the preprint by Gorroño et al. https://egusphere.copernicus.org/preprints/2025/egusphere-2025-4924/, which is focused on this topic.

Thank you for this suggestion! We were not aware of this paper as it was submitted around the same time as ours. We have expanded the discussion with points raised by Gorroño et al. (2025):

"In this context, a recent study by Gorroño et al., (2025) systematically investigated the effect of different observation and illumination geometries on the retrieved $CH_4$ maps (i.e. parallax effect) and the resulting emission estimates. They showed that large VZAs and SZAs can lead to artificial elongation or compression of plumes along the plume direction. This bias in apparent plume length L directly propagates into emission estimates and likely also occurred in the observations analysed in this study. However, their influence is probably masked by the comparatively large variability in wind speed.

Furthermore, Gorroño et al., (2025) found that the parallax effect substantially reduces the PoD due to lower apparent $CH_4$ enhancements. In their simulations, the PoD varied between approximately 0.5 and 0.8 depending on the angular configuration. For the present study, the influence of parallax effects is likely minor, as the detection outcomes shown in Figure 6 are primarily controlled by wind speed and flight altitude. The few non-detected plumes with emission rates exceeding 5 kg $CH_4$ $h^{-1}$ at low wind speeds are instead attributable to overlaps with retrieval artefacts.

Gorroño et al., (2025) also demonstrated that when the effective wind speed $U_{eff}$ is calibrated against the 10m wind speed $U_{10}$ using L, biases in L translate into systematic errors in the calibration itself. As a consequence, emission estimates exhibit errors below 10% for mid-latitude summer conditions, but can reach up to 30% for wintertime observations. In the context of this study, the parallax-induced bias in $U_{eff}$ is only relevant for emission estimates derived using the $U_{eff}$ parametrisation of Varon et al., (2018) and does not affect estimates based on wind speeds at the source height. To mitigate the effect of viewing geometry, Gorroño et al., (2025) recommended to explicitly account for observation and illumination geometry in the planning of flight paths

for airborne sensors and to calibrate U$_{eff}$ using plume simulations that match the angular configuration ("train as you measure").

Overall, additional work is needed to correct for the parallax effect, especially as this phenomenon also affects instruments with coarser spatial resolution even if they do not spatially resolve the plume shadow (Schwaerzel et al., 2020)."

2. AVIRIS-4-specific Ueff model (section 2.5 and 3.4): the authors test the Ueff model developed by Varon et al. for GHGSat. Even if they find that this model combined with the 10-m lidar wind speed helps reduce the offset between the reported and estimated Qs, I wonder if an AVIRIS-4-specific Ueff model should be used for this test? This could be one model trained for the specific conditions of these acquisitions, at least in terms of retrieval noise and range of emissions. Regarding the spatial resolution, I acknowledge it is not feasible to recreate the very high spatial resolution of the AVIRIS-4 observations in the training of the model, but even getting to a 25-m sampling might help further improve the results.

We fully agree that an AVIRIS-4 specific U$_{eff}$ model could be insightful to explore the impact of the effective wind speed on emission estimates. However, this requires expensive Large-Eddy (LES) or Direct Numerical (DNS) simulations at meter scale resolution comparable to the ground pixel size of AVIRIS-4. We are indeed working on such simulations and will present the results in a forthcoming study. Including such simulation results would be well beyond the scope of the present study.

We use the U$_{eff}$ of 1.47 x u10 estimated by Varon et al. for the CSF method. However, this U$_{eff}$ was estimated for plumes detected by GHGSat, which are longer than those observed in this study and have therefore experienced more vertical mixing. As a consequence, their effective wind is likely overestimated for the comparatively short AVIRIS-4 plumes, which is indeed what we find in our analysis.

- MF vs LMF (sections 2.4.1, 2.4.2 and 3.5.5): the basic and log versions of the matched filter retrieval are compared. The authors find that "improved detectability can be attributed (...) to reduced noise levels in the CH4 retrievals," (by the LMF), and that "the LMF had little to no effect on CH4 enhancements for the largest release events in the campaign". This is actually interesting, as I would expect the opposite results: the LMF leading to a worse plume detectability due to the higher sensitivity to the surface and to it being more prone to generate false positives, but at least being helpful to correct the underestimation of XCH4 in the stronger emissions. Could you please comment?

We agree that the LMF increases sensitivity to low-SNR pixels and surface effects, which can lead to larger systematic biases in background CH$_4$ values (see answer to question 2 of reviewer 1). With "(...) reduced noise levels in the CH4 retrievals (...)" we specifically mean reduced random, small-scale variability within regions of similar surface properties, not reduced bias. In our dataset, we observe that although the LMF output can exhibit larger background offsets, the pixel-to-pixel variability within homogeneous albedo patches is in some cases lower than for the MF. This is illustrated in the upper row of Figure 15, where the background around the release location appears smoother for the LMF, making the plume more distinguishable.

One possible explanation is that background variability in the radiance measurements is predominantly multiplicative. Applying a logarithmic transformation converts multiplicative variations into additive ones, potentially yielding a background covariance structure that is closer to the Gaussian assumptions used in the matched filter. This could lead to a smoother background and reduced apparent random noise in the

retrieved CH$_4$ maps. However, further research is needed to examine the exact reasons, which would be beyond the scope of this study.

We have revised the mentioned section as follows to clarify the distinction between offset and random noise:

"This improved detectability can be attributed, in part, to reduced random background variability in the retrieved CH4 maps, which facilitated more confident identification of the plume signal. However, the LMF also introduces larger systematic biases in background CH4 values compared to the MF, as evident in both the upper and lower rows of Figure 15. An analysis of the eigenvalues of the covariance matrices for different surface albedos suggests that these biases are associated with increased sensitivity of the log-transformed radiances to pixels with low SNR, which is the case for albedo surfaces with low albedo."

For large release events, our iterative MF already compensates for most of the non-linear absorption associated with high optical depths (see answer to question 1 of reviewer 1). Consequently, for the strongest plumes the MF already reproduces the forward model sufficiently well, leaving little room for improvement by the LMF. We have added a respective paragraph to the discussion:

"On the other hand, the LMF increased local biases in the retrieved CH$_4$ maps which we attribute to the amplification of noise by the log-transform in pixels with low SNR, caused by low albedo. This spatially more heterogeneous background can obscure small enhancements or produce false detections. In contrast to Schaum (2021), we did not observe an improved performance of the LMF for large release events. The iterative MF applied in our study seems to successfully account for most non-linear absorption in pixels with large CH$_4$ enhancement."

- POD (section 2.7): I agree that this type of controlled release experiment including multiple overpasses can be very useful to establish POD functions for methane-sensitive instruments. However, I don't understand the relationship of this section with the rest of the study. Is Eq. 21 a result from the analysis (hence it should be moved to Sec 3)? how is it related to the plume detection limits discussed in other parts of the study? Would it be possible to show the dependence of the detection limits on U10 based on this equation?

We agree that Eq. 21 better fits into section 3 and have therefore moved it.

We have clarified that Figure 6 in our manuscript shows the dependence of the detection limit based on Eq. 21 in the revised manuscript:

"(a) Reported CH4 emissions vs. on-site lidar wind measurement at 10 m. (b) Probability of detection for a flight altitude of 1000 m. above mean sea level using Eq. 21."

- Estimation of CH4 retrieval uncertainty: Eq. 13 is set to account for measurement errors propagated to retrieval errors, but wouldn't it be better to estimate the 1-sigma retrieval error from the data themselves, e.g. as the StdDev of XCH4 over plume-free regions of the scene?

In this study, we chose to model the uncertainties of the retrieved $CH_4$ columns $\sigma_V$ as specified in Eq. 13 to have a per-pixel uncertainty which accounts for local variations in albedo. Such a per-pixel uncertainty is required for the Gaussian fits for the CSF.

[Figure]

**Figure 5: Comparison of $\sigma_V$ from propagation of uncertainties according to Eq. 13 and from data.**

We compared $\sigma_V$ computed using Eq. 13 with $\sigma_V$ estimated directly from the observations as the standard deviation in a plume free region next to the release site with similar surface properties. In Figure 5, one can see that the $\sigma_V$ derived from the data is approximately one order of magnitude larger than the $\sigma_V$ derived from the propagation of uncertainty. This implies that our current noise representation is incomplete. One reason for this could be the fact that we had to rely on the EMIT noise model as the AVIRIS-4 noise model is still under development.

Therefore, we have switched to the method suggested by the reviewer for calculating $\sigma_V$ and adjusted the corresponding section in the methods section:

"(…) where $\sigma_{CH_4}$ is the standard deviation of retrieved $CH_4$ columns in a plume-free region next to the release location with similar surface properties."

To assess the sensitivity of the estimated emissions to $\sigma_V$ we increased this parameter by a factor of 10 and re-calculated the emission estimates. This change had no effect on the mean estimated $CH_4$ emissions (+1.5e-07%) and only a negligible influence on the average precision of the retrieved $CH_4$ emissions (+2.5%). This insensitivity arises because uniformly larger uncertainties in the $CH_4$ columns (used as weights for the Gaussian fits) have a minor influence on the integrated $CH_4$ mass.

3.  Estimation of emission rates: both the IME and the CSF models are used, and one or the other are being selected depending on the plume morphology ("L300, We were able to estimate the emission from 67 of the 68 detected plumes, 54 of which were estimated using the CSF method and 13 using the IME method"). How is the method actually determined for each plume? Related to that, in L280 the authors write "the CSF method outperforms the IME approach, as the effect of turbulence is reduced through averaging across multiple cross-sections". This is an important statement in my opinion, and it would be nice to see it more deeply discussed. Also, I think it would be very interesting to see a comparison of the Qs between the two methods for the plumes, both between each other and with the metered values.

We acknowledge that the description, under which circumstances we use the CSF or IME, was a bit vague. We have therefore adjusted it on line 161ff:

"We used the CSF for longer plumes and more turbulent conditions as it averages the fluxes along several cross-sections. Conversely, the IME was used for short plumes and plumes that deviate from a Gaussian plume shape such as for overlapping double plumes."

As suggested, we also estimated the $CH_4$ emissions using the IME for the cases where we used the CSF before. The opposite was not possible as the CSF cannot be applied to the plumes which do not follow a Gaussian distribution (e.g. overlapping double plumes etc.). This is the reason why we used the IME for these cases in the first place. Consequently, Figure 6 shows the estimate emissions from 53 out of 68 observed plumes.

[Figure]

**Figure 6: Comparison of emission estimates using the CSF and IME.**

Subplot (a) shows that the relative errors in estimated $CH_4$ emissions are smaller when using the CSF (median: 24.1%) compared to the IME (median: 34.3%) and that the latter shows a larger spread. As mentioned in the methods section of the paper, this is due to the fact that the CSF is more robust against turbulence than the IME. When comparing the absolute estimated emissions in subplot (b), it becomes evident that the IME tends to result in lower emission estimates than the CSF for small release events. This likely happens because background noise has a larger influence on the integrated $CH_4$ mass than when using the CSF due to the small integration area. For larger release events, the estimated emissions are higher when using the IME compared to the CSF. This probably occurs because the CSF can better account for local variations of the background by fitting a linear background with the Gaussian curve for each sub polygon (see Kuhlmann et al., 2024).

We have also added this figure and description to the supplement.

4.  Estimation of emission rates without explicit use of external U10 data: one of the main conclusions of this study is that accurate wind speed information is key for accurate emission rate estimates (e.g. sections 4.2.1 and 4.2.2). I am wondering, does this call for the development of ML-based methods able to infer emission rates solely from 2D XCH4 plume maps, without specific use of external U10 information (e.g. https://www.sciencedirect.com/science/article/abs/pii/S0034425721005290, https://egusphere.copernicus.org/preprints/2025/egusphere-2025-1075/)? This could be discussed in these sections

Our results indeed show that uncertainties in the effective wind speed are disproportionately large, and that methods that do not rely on external wind information represent a promising way to reduce this dominant

error source (L 525ff). In addition to the observation-based approaches already discussed, recent machine-learning models offer the same advantage. We were aware of the studies cited by the reviewer and agree that such methods are highly relevant for future developments. In the revised manuscript, we have expanded Section 4.2.2 to include a brief discussion of these ML-based approaches, their potential as well as their current limitations:

"Alternatively, machine learning based models could be used to estimate trace gas emissions either directly from radiance data (e.g Joyce et al., 2023, Rouet et al., 2024) or from plume images (e.g. Jongaramrungruang et al., 2022, Bruno et al., 2024, Ouerghi et al., 2025, Plewa et al., 2025). These approaches have recently shown that it is possible to infer emission rates without explicitly relying on external wind data. Their main advantages are that they can, just as the other approach outlined above, bypass wind speed uncertainties and additionally, provide rapid and automated emission estimates at large scales.

While these models are very promising, they are still limited in their representativeness due to a lack of wind speed information within a single image. Furthermore, they provide limited interpretability and their uncertainty quantification is still less mature than for the traditional approaches based on the mass balance."

Other minor comments:

-   L1: what is a "critical greenhouse gas"?

Adjusted to "potent"

-   L2: I would replace "use" by "sector"

Replaced by "sector"

-   L34: CO2M is not yet flying, unlike the rest of the missions mentioned in this paragraph.

We removed "CO2M" from the list.

-   L101: "Observations over dark surfaces (...) have a low SNR

Adjusted accordingly

-   Fig. 11: could the pixel size of each flight altitude be added?

We have added the pixel size for each flight altitude.

-   L428 "We assume that this technique can be applied if the length of the shadow is at least twice as large as the uncertainty in the length of the shadow". I don't understand what the basis for this statement is.

Our original wording did not sufficiently justify this criterion. The requirement that the shadow length exceeds twice its uncertainty is intended as a practical detectability condition rather than a strict statistical confidence threshold.

The uncertainty in the shadow length arises primarily from pixel discretization at the shadow boundaries, which can contribute up to one mixed pixel at both the upper and lower edge of the shadow. Requiring the shadow length to exceed this uncertainty by at least a factor of two ensures that the measured shadow is clearly resolvable.

To clarify this, we have adjusted the corresponding sentence:

"We assume that this technique can be reliably applied only if the measured shadow length exceeds its measurement uncertainty by a sufficient margin. The uncertainty in the shadow length is dominated by pixel

discretization at the shadow boundaries, where at most one mixed pixel can occur at both the upper and lower edge of the shadow. Requiring the shadow length to be at least twice this uncertainty ensures that the shadow is sufficiently resolved. Under this criterion, the minimum emission height that can be resolved is given by (...)"

- L461-465: I think this type of information about AVIRIS-4 should be included in the Introduction section

We moved this sentence to the introduction section.

- L483: "the linearisation of the unit absorption spectrum no longer holds and assumed enhancements for the calculation of the absorption spectrum have greater influences on the retrieved enhancements" – has this been shown in this study?

We did not test this systematically but explored the effect of different a-priori $CH_4$ enhancements for the calculation of the unit absorption spectrum when implementing our iterative approach. In the revised manuscript, we have added the specification "(...) linearisation of the unit absorption spectrum around $\alpha = 0$ no longer holds (...)". Additionally, we have added a few sentences emphasising the need of an iterative approach:

In the methods section: "Our iterative approach reduces the approximation error introduced by the linearisation of the Beer-Lambert law by expanding around the current estimate of $\alpha$ rather than $\alpha = 0$, which decreases the linearisation error quadratically in the update step. For large enhancements, this substantially mitigates non-linear absorption effects"

In the results section: "This is likely because our iterative MF already compensates for most of the non-linear absorption associated with high optical depths"

In the discussion: "Instead, the iterative MF already accounted for most non-linear absorption in pixels with large CH4 enhancement, while the LMF's susceptibility to noise amplification limited its utility in areas with lower SNR."

**References**

Gorroño, J., Pei, Z., Valverde, A., and Guanter, L.: Considering the observation and illumination angular configuration for an improved detection and quantification of methane emissions, EGUsphere [preprint], https://doi.org/10.5194/egusphere-2025-4924, 2025.

Kuhlmann, G., Koene, E., Meier, S., Santaren, D., Broquet, G., Chevallier, F., Hakkarainen, J., Nurmela, J., Amorós, L., Tamminen, J., and Brunner, D.: The *ddeq* Python library for point source quantification from remote sensing images (version 1.0), Geosci. Model Dev., 17, 4773–4789, https://doi.org/10.5194/gmd-17-4773-2024, 2024.